# Metabolic sensing in AgRP neurons integrates homeostatic state with dopamine signalling in the striatum

Alex Reichenbach[1], Rachel E Clarke[1], Romana Stark[1], Sarah Haas Lockie[1], Mathieu Mequinion[1], Harry Dempsey[1], Sasha Rawlinson[1], Felicia Reed[1], Tara Sepehrizadeh[2], Michael DeVeer[2], Astrid C Munder[1,3], Juan Nunez-Iglesias[4], David C Spanswick[1,5], Randall Mynatt[6], Alexxai V Kravitz[7], Christopher V Dayas[8], Robyn Brown[3,9], Zane B Andrews[1]*

[1]Monash Biomedicine Discovery Institute and Department of Physiology, Monash University, Clayton, Australia; [2]Monash Biomedical Imaging Facility, Monash University, Clayton, Australia; [3]Florey Institute of Neuroscience & Mental Health, Parkville, Australia; [4]Monash Biomedicine Discovery Institute and Department of Anatomy and Developmental Biology, Monash University, Clayton, Australia; [5]Warwick Medical School, University of Warwick, Coventry, United Kingdom; [6]Gene Nutrient Interactions Laboratory, Pennington Biomedical Research Center, Louisiana State University System, Baton Rouge,, United States; [7]Departments of Psychiatry, Anesthesiology, and Neuroscience, Washington University in St Louis, St Louis, United States; [8]School of Biomedical Sciences and Pharmacy, University of Newcastle, Newcastle, Australia; [9]Department of Biochemistry and Pharmacology, University of Melbourne, Parkville, Australia

*For correspondence:
zane.andrews@monash.edu

Competing interest: The authors declare that no competing interests exist.

**Abstract** Agouti-related peptide (AgRP) neurons increase motivation for food, however, whether metabolic sensing of homeostatic state in AgRP neurons potentiates motivation by interacting with dopamine reward systems is unexplored. As a model of impaired metabolic-sensing, we used the AgRP-specific deletion of carnitine acetyltransferase (*Crat*) in mice. We hypothesised that metabolic sensing in AgRP neurons is required to increase motivation for food reward by modulating accumbal or striatal dopamine release. Studies confirmed that *Crat* deletion in AgRP neurons (KO) impaired ex vivo glucose-sensing, as well as in vivo responses to peripheral glucose injection or repeated palatable food presentation and consumption. Impaired metabolic-sensing in AgRP neurons reduced acute dopamine release (seconds) to palatable food consumption and during operant responding, as assessed by GRAB-DA photometry in the nucleus accumbens, but not the dorsal striatum. Impaired metabolic-sensing in AgRP neurons suppressed radiolabelled 18F-fDOPA accumulation after ~30 min in the dorsal striatum but not the nucleus accumbens. Impaired metabolic sensing in AgRP neurons suppressed motivated operant responding for sucrose rewards during fasting. Thus, metabolic-sensing in AgRP neurons is required for the appropriate temporal integration and transmission of homeostatic hunger-sensing to dopamine signalling in the striatum.

## Editor's evaluation

This manuscript will be of broad interest to behavioral neuroscientists studying energy homeostasis, hypothalamic feeding circuits, and dopamine. The paper uses a genetic mouse model to study critical connections between homeostatic circuitry and dopamine release in response to food reward.

The experimental results support key claims of the paper, and tie in nicely with previously published data.

## Introduction

The motivation to approach and consume food depends not only on the palatability and caloric density of the available food source, but also on the energy state of the organism. For example, when food is abundant many prey species forage within known territories to reduce survival threats (*Sih, 1980*). Conversely, when food is scarce, animals are motivated to take greater risks and forage within unfamiliar territories to search for food (*Whitham and Mathis, 2000*). Thus, the motivation to seek palatable, energy-dense food evolved as a key mechanism for survival and maturation in an environment with limited food resources. Given that heightened motivation for palatable food in an environment of low food availability shaped an evolutionary benefit, it is not surprising that homeostatic feeding circuits can have a profound effect on motivation. With high caloric foods readily available, these circuits may contribute to the overconsumption of highly palatable and calorie dense foods, which is a leading cause for today's obesity crisis. Indeed, human evidence shows that fasting biases reward systems to high caloric foods (*Goldstone et al., 2009*; *Cameron et al., 2014*).

Agouti-related peptide (AgRP) neurons in the arcuate nucleus of the hypothalamus (ARC) are a critical population of hunger-sensitive neurons that primarily function to increase appetite or conserve energy (*Aponte et al., 2011*; *Gropp et al., 2005*; *Krashes et al., 2011*; *Luquet et al., 2005*; *Ruan et al., 2014*; *Siemian et al., 2021*). A key element of appetitive behaviour is the increased motivation for goal-directed outcomes. Intriguingly, hunger has been used for decades in behavioural neuroscience to improve performance and learning in operant tasks (*Ramond, 1954*). AgRP neurons function within this framework by increasing the willingness to work for food and food rewards to the same level as that seen in fasted mice (*Krashes et al., 2011*; *Atasoy et al., 2012*; *Betley et al., 2015*; *Chen et al., 2016*). Moreover, AgRP neurons influence motivation circuits (*Alhadeff et al., 2019*; *Dietrich et al., 2012*; *Mazzone et al., 2020*), with chemogenetic activation elevating dopamine release in response to food consumption (*Alhadeff et al., 2019*). Since AgRP neurons are most active during periods of energy deficit (*Baskin et al., 1999*; *Briggs et al., 2011*; *Mandelblat-Cerf et al., 2015*; *Schwartz et al., 1998*; *Yang et al., 2011*), we hypothesised that they transmit hunger-specific metabolic information to motivation circuits. However, whether AgRP neurons are required to gauge homeostatic state through metabolic sensing, to influence motivation neural circuits, remains unknown.

Sensory cues that predict food availability and palatability rapidly suppress hunger-sensitive AgRP neuronal activity in hungry mice (*Betley et al., 2015*; *Mandelblat-Cerf et al., 2015*; *Chen et al., 2015*; *Su et al., 2017*). Although this decrease in activity occurs prior to food consumption, post-ingestive caloric feedback is required to sustainably reduce AgRP neuronal activity (*Su et al., 2017*; *Beutler et al., 2017*). Indeed, the reduction of AgRP neuronal activity correlates with the number of calories ingested and was not observed after repeated ingestion of a non-caloric sweetened gel (*Su et al., 2017*). These studies highlight that the maintenance of normal AgRP activity in response to post-ingestive gastrointestinal feedback requires metabolic sensing of available calories in combination with gut-brain neural communication (*Su et al., 2017*; *Beutler et al., 2017*; *Goldstein et al., 2021*).

Previously, we demonstrated that carnitine acetyltransferase (*Crat*) in AgRP neurons was required as a molecular sensor for peripheral substrate utilisation during fasting and refeeding (*Reichenbach et al., 2018c*). Moreover, *Crat* activity in AgRP neurons programmed a broad metabolic response of the AgRP proteome and was required to promote normal refeeding after fasting (*Reichenbach et al., 2018c*). The reduced feeding response after fasting in AgRP *Crat* KO mice suggested that metabolic sensing of homeostatic state by AgRP neurons transmits hunger-specific metabolic information into neural circuits controlling dopamine signalling and the motivational aspects of food-directed behaviour. Here, we describe experiments demonstrating that impaired metabolic sensing in AgRP neurons, using the conditional deletion of *Crat* from AgRP neurons as a validated model, reduces motivated behaviour during an operant task and dopamine release in the nucleus accumbens (NAc) or dorsal striatum, over different time frames, in response to palatable food rewards. These studies highlight that metabolic-sensing in AgRP neurons is required for the appropriate temporal integration

of hunger-sensing to potentiate food reward-related dopamine release in the striatum and control motivation for palatable food rewards.

## Results

### *Crat* deletion in AgRP neurons is a valid model of impaired metabolic sensing

To demonstrate that *Crat* deletion in AgRP neurons is a reliable model of impaired metabolic sensing, we prepared hypothalamic brain slices from WT and KO mice for electrophysiological characterisation of glucose sensing in AgRP neurons. We detected no differences in the fundamental electrophysiological properties of AgRP neurons including resting membrane potential (*Figure 1A*), input resistance (*Figure 1B*), spontaneous firing frequency (*Figure 1C*), although peak action potential amplitude was significantly lower in KO mice (*Figure 1D*). Collectively, these studies demonstrate that *Crat* in AgRP neurons has little effect on the intrinsic electrophysiological properties of the cell. Similarly, no genotype-dependent changes in the frequency of spontaneous excitatory or inhibitory post-synaptic potentials (EPSPs or IPSPs, respectively) were detected (*Figure 1—figure supplement 1A, C*), indicating no differences in the external synaptic input onto AgRP neurons in WT compared to KO mice. To show impaired glucose-sensing in AgRP neurons, we recorded neuronal responses to an increase in extracellular glucose from a basal level of 2 mM to 5 mM. These glucose concentrations were chosen to represent brain glucose concentrations under fasting and fed conditions, based on estimates in the CSF from fasted and fed rats and mice (*van den Top et al., 2017*). Glucose-excited neurons were defined based upon a response characterised by membrane potential depolarisation and/or an increase in action potential firing frequency with increased extracellular glucose (*Figure 1—figure supplement 1B*). Glucose-inhibited cells were identified by responses to increased glucose characterised by membrane potential hyperpolarisation and/or a reduction in spontaneous action potential firing frequency (*Figure 1—figure supplement 1D*). In WT mice, 48% of cells (n = 10/21) were excited, 33% of cells (n = 7/21) were inhibited and 19% (n = 4/21) were insensitive to changes in extracellular glucose (*Figure 1E*). In KO mice, the number of glucose-excited cells was reduced compared to WT, with only 28% classified as glucose-excited (n = 7/25). Similarly, the incidence of glucose-inhibited cells was reduced to 16% (n = 4/25) of the population compared to WT mice, with the majority of cells not responding to 5 mM glucose 56% (n = 14/25) and classified as glucose insensitive (*Figure 1F*). In total 17/21 AgRP neurons responded to an increase in extracellular glucose in WT mice, whereas as only 11/25 AgRP neurons responded to glucose in KO mice, representing a decrease in glucose-responsive neurons (WT 81% vs KO 44%). To examine whether these ex vivo recordings translated to in vivo changes, we injected a cre-dependent GCAMP6s-expressing virus into *Agrp*$^{cre/wt}$ mice to record AgRP neural activity in freely moving WT and KO mice. An increase in activity to ghrelin was used as an index of correct viral expression and fibre optic placement. Only mice with a maximal peak z-score of >4 were included for analysis in experimental groups, using this criterion 5/19 mice, across both WT and KO mice, were excluded for experimentation (*Figure 1—figure supplement 1N-O*). Fed or overnight fasted WT and KO mice were injected with glucose (2.25 g/kg) and GCaMP6s fluorescence from AgRP neurons was measured for 10 min post injection (*Figure 1—figure supplement 1J*, K, M). Glucose injection suppressed AgRP activity in WT and KO mice; however, activity quickly returned to baseline levels in fasted WT but not KO mice (*Figure 1—figure supplement 1L*). Therefore, despite identical infusion of glucose, the activity of AgRP neurons in KO mice failed to rebound, which could dampen the motivation to seek food.

To examine these differences in AgRP response in a more physiological relevant scenario, fed WT or KO mice were presented with a novel peanut butter chip (PB) and the fall in AgRP activity was measured, as previously described (*Betley et al., 2015*; *Mandelblat-Cerf et al., 2015*; *Chen et al., 2015*). The fall in AgRP activity was similar in WT and KO mice in response to first PB exposure (*Figure 1G–H*), however, with repeated exposure across different sessions WT mice showed a greater suppression of activity relative to first exposure (*Figure 1—figure supplement 1E*; *Figure 1L*, *Video 1*), consistent with previous studies showing that the magnitude of activity suppression is proportional to calories consumed (*Su et al., 2017*; *Beutler et al., 2017*). Unlike WT mice, KO mice did not exhibit any further reduction in AgRP activity to repeated PB exposure (*Figure 1—figure supplement 1F*, *Figure 1L*, *Video 2*) and the reduction in AgRP neural activity in KO mice to repeated

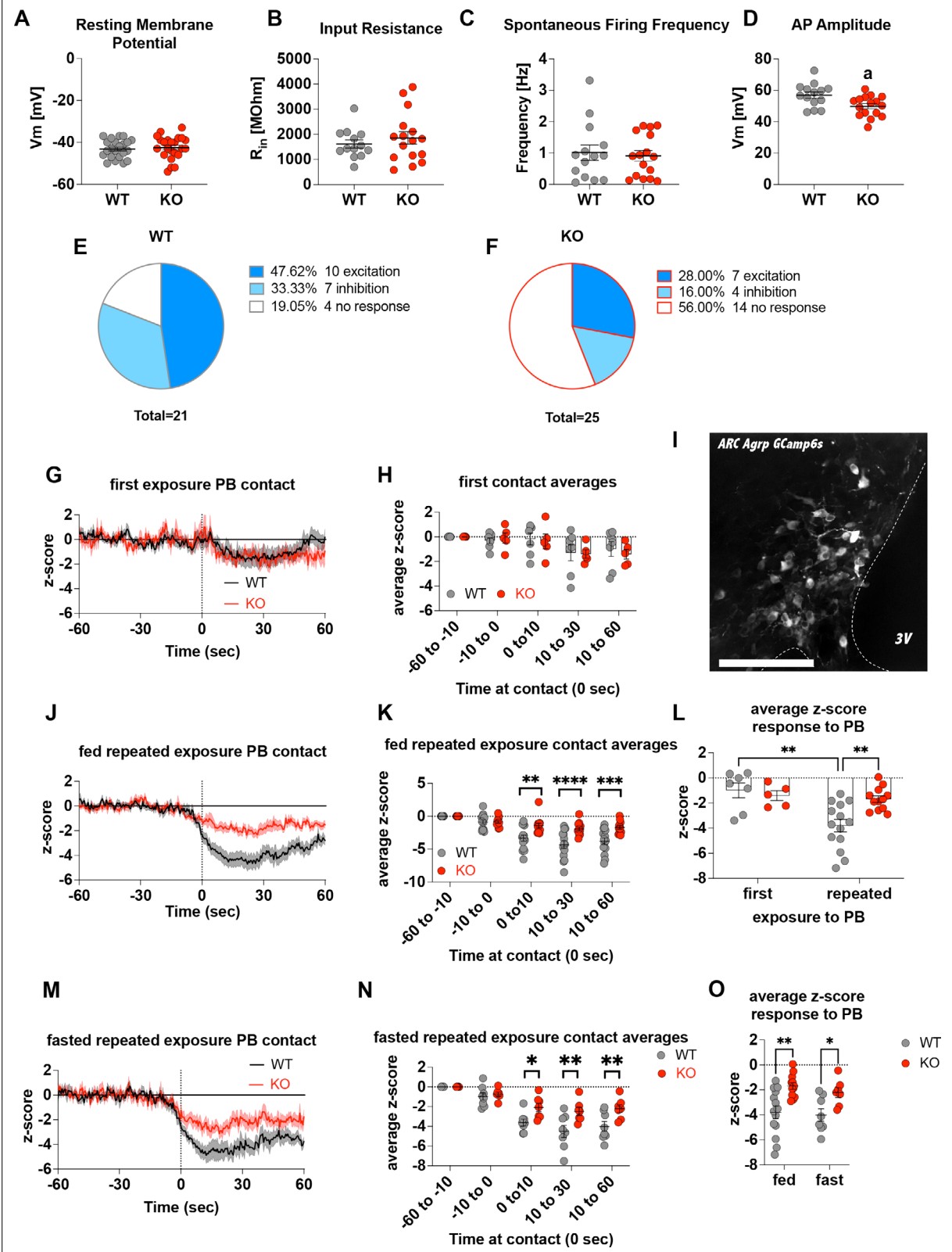

**Figure 1.** Deletion of Crat in AgRP neurons is a model of impaired metabolic sensing. Intrinsic electrophysiological properties were little affected in WT versus KO mice including resting membrane potential (**A**, n = 22/23 cells WT/KO), input resistance (**B** n = 13/16 cells WT/KO), spontaneous action potential firing frequency (**C** n = 14/16 cells WT/KO), although action potential amplitude was significantly reduced in KO AgRP neurons (**D**, n = 14/16 cells WT/KO). Glucose-responsive profiles of AgRP neurons from WT (21 cells) and KO (25 cells) mice characterised by their response to increased

*Figure 1 continued on next page*

*Figure 1 continued*

extracellular glucose concentration from 2 mM to 5 mM (**E–F**). In vivo fiber photometry analysis of AgRP neuronal activity in WT and KO mice (**G–O**). First exposure to a peanut butter chip (PB) (**G**, n = 7 WT; n = 5 KO) and time binned summary data (**H**). GCaMP6s expression in AgRP neurons [scale bar = 100 µm] (**I**). AgRP neural activity in ad libitum-fed mice in response to repeated PB exposure (**J**; n = 14 WT; n = 12 KO) and time binned summary data (**K–L**). AgRP neural activity in fasted mice in response to repeated PB exposure (**M**; n = 8 WT; n = 7 KO) and time binned summary data (**N**), n = 8 WT, n = 7 KO). Time binned summary data comparing the response to repeated PB exposure in ad libitum-fed or fasted WT and KO mice (**O**). Data± SEM; ANOVA with Tukey's post hoc analysis (**H, K, L, N, O**) and unpaired students t-test (**A–D**); a, significant at p < 0.05; ** p < 0.01, *** p < 0.001, **** p < 0.0001. Dashed lines in I, J, and M indicated the moment of contact with a PB contact. *Videos 1 and 2* accompany this figure.

The online version of this article includes the following source data and figure supplement(s) for figure 1:

**Source data 1.** Deletion of Crat in AgRP neurons is a model of impaired metabolic sensing.

**Figure supplement 1.** Deletion of Crat in AgRP neurons is a model of impaired metabolic sensing.

**Figure supplement 1—source data 1.** Deletion of Crat in AgRP neurons is a model of impaired metabolic sensing.

**Figure supplement 2.** Deletion of Crat from AgRP neurons does not affect ghrelin-induced food intake or AgRP activation.

**Figure supplement 2—source data 1.** Deletion of Crat from AgRP neurons does not affect ghrelin-induced food intake or AgRP activation.

PB exposure was significantly attenuated compared to WT mice at all time points after PB exposure either in ad libitum-fed (*Figure 1J–K*) or fasted mice (*Figure 1M–N*). These results suggest KO mice could not integrate caloric information associated with PB consumption either in the fed or fasted state and therefore did not show the expected greater reduction in activity. To determine if this response was specific to metabolic sensing of calorie content and not a generalised response due to impaired AgRP neuronal function caused by *Crat* deletion, we examined ghrelin-induced food intake and AgRP neural activity (*Figure 1—figure supplement 2*). IP injection of ghrelin (1 mg/kg) significantly increased food intake within 4 hr (*Figure 1—figure supplement 2B*; main effect of ghrelin); however, no differences in genotype were observed. IP ghrelin significantly increased AgRP neural activity (*Figure 1—figure supplement 2D-E*; main effect of ghrelin); however, no differences in genotype were observed and the presentation of chow diet to ghrelin-injected mice produced an equal suppression in AgRP neural activity (*Figure 1—figure supplement 2F-G*; main effect of food access), again without differences between WT and KO mice. Finally, we observed no differences in anxiety-like behaviour in the light-dark box or elevated-plus maze (*Figure 1—figure supplement 2I-P*), although KO mice showed reduced locomotor activity in the elevated-plus maze but not light-dark box. The lack of genotype differences in ghrelin-induced food intake and AgRP activity, or differences in anxiety-like behaviour, which can be modulated by AgRP neurons (*Dietrich et al., 2015*; *Padilla et al., 2016*), suggests the deletion of *Crat* from AgRP neurons does not cause a global deficit in function. Taken together, the specific impairments in ex vivo glucose sensing and in vivo responding to IP glucose or palatable calorie consumption

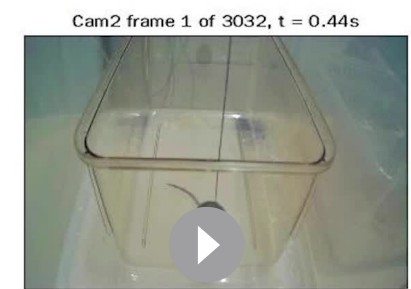

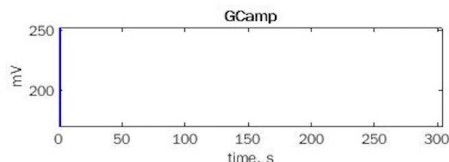

**Video 1.** GCaMP6s AgRP neural activity time locked to behaviour at 10 x normal speed in an ad libitum-fed WT mouse previously exposed to a peanut butter pellet. The video shows raw data collected in mV at the photoreceiver prior to df/f calculations.

https://elifesciences.org/articles/72668/figures#video1

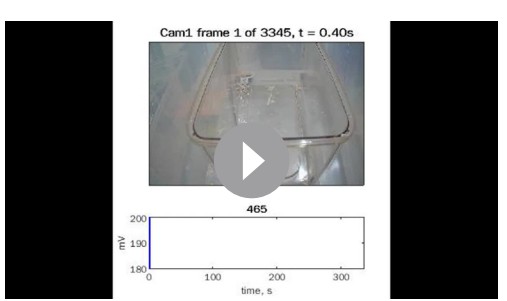

**Video 2.** GCaMP6s AgRP neural activity time locked to behaviour at 10 x normal speed in an ad libitum-fed KO mouse previously exposed to a peanut butter pellet. The video shows raw data collected in mV at the photoreceiver prior to df/f calculations.

https://elifesciences.org/articles/72668/figures#video2

show that *Crat* deletion in AgRP neurons is a useful model to assess the impact of impaired metabolic sensing in AgRP neurons on motivated behaviour and dopamine release in response to food rewards.

## Metabolic sensing in AgRP neurons affects mesolimbic dopamine pathways

Previous studies show that chemogenetic activation of AgRP neurons modulates dopamine release in the NAc to food (*Alhadeff et al., 2019*). To assess whether impaired metabolic sensing in AgRP neurons affected dopamine release in the NAc, we used GRAB-DA sensors with in vivo fiber photometry (*Sun et al., 2018*). Acute dopamine release in the NAc was measured in response to a non-food object (wood dowel), chow food and a peanut butter (PB) chip with each presentation separated by 2 min (presented in that order) (*Figure 2A*). We observed a significant increase in dopamine release in response to chow or a PB chip presentation when compared to wood dowel in both fed and fasted WT mice 0–30 s after placing the food or object into the cage (*Figure 2D, E, H1*; *Videos 3–6*). Strikingly, KO mice had significant lower NAc dopamine release between 0 and 30 s, as assessed by average z-score, after a non-food object or PB chip was placed in the cage compared to WT fed mice (*Figure 2F*). Average z-score was also significantly reduced in fasted KO mice in response to chow and PB chip (0–30 s) when compared to WT fasted mice (*Figure 2J*). At 30–60 s after food or an object was placed in the cage, the average z-score was only significantly lower in KO in response to PB in the fed mice (*Figure 2G and K*), suggesting most of the differential effects occur within the first 30 s after presentation.

Furthermore, we aligned the dopamine signal to first contact with the PB chip (*Figure 2L and M*) and showed that there was a significantly reduced dopamine response 0–30 s after contact with PB in KO compared to WT mice in both fed and fasted states (*Figure 2O and P*). No differences in dopamine release were observed in the 20 s prior to first contact or from 30 to 60 s after contact (*Figure 2O and P*). Taken together, these results highlight that impaired metabolic sensing in AgRP neurons affects acute dopamine release in the NAc in both fed and fasted states in response to chow and a peanut butter chip. Time spent eating or time to first contact were not significantly different in WT and KO mice in fed or fasted state showing that these factors could not account for genotype differences in dopamine release in the NAc, although there was an expected main effect for fasting to increase overall time spent eating compared to fed mice, irrespective of genotype (*Figure 2Q and R*).

## Metabolic sensing in AgRP neurons affects motivated behaviour

Most operant conditioning protocols are conducted in stand-alone operant chambers and typically include a mild calorie restriction to facilitate learning the operant task. This exploits the fact that food restriction enhances appetitive drive, which is a result of elevated AgRP neuronal activity (*Krashes et al., 2011*; *Chen et al., 2016*). To avoid this potential confound we placed the open source Feeding Experimental Device 3 (FED3) (*Matikainen-Ankney et al., 2021*) in home cages to facilitate learning in a low stress environment for long periods of time. As such, operant learning can be acquired quickly and easily in ad libitum-fed mice without the need for caloric restriction – an important consideration for our experiments in which differential feeding responses to fasting, as reported previously (*Reichenbach et al., 2018c*; *Reichenbach et al., 2018b*), may affect task acquisition and performance (*Figure 3A*). With this approach, there were no differences in operant responding during FR sessions (*Figure 3C*), indicating no differences in learning the operant task. To test motivation, progressive ratio sessions were performed overnight with (fed) and without access (fasted) to chow. Importantly, in fasted PR experiments, mice began fasting at the same time FED3s were placed in home cages (*Figure 3B*). This was designed to test the motivational response to an increasing energy deficit, which requires an awareness of homeostatic state over time, rather than in response to an existing energy deficit after a prolonged fast. During PR fed experiments, WT and KO mice displayed a similar breakpoint (*Figure 3D*) and total number of pellets collected (*Figure 3E*), however, during a fasted PR, KO mice showed a reduced breakpoint and total pellet consumption compared to WT mice (*Figure 3D and E*). Active (*Figure 3F*) but not inactive nose pokes (*Figure 3G*) during the fasted PR session were significantly lower in KO mice compared to WT, as well as pellets received during the progressive ratio session (*Figure 3H*). Intriguingly, active nose pokes and pellets collected only diverge around 6–8 hr after the beginning of fasting and the PR session, similar to when food intake diverges after fasting as previously described (*Reichenbach et al., 2018c*; *Reichenbach et al., 2018b*). This timepoint likely

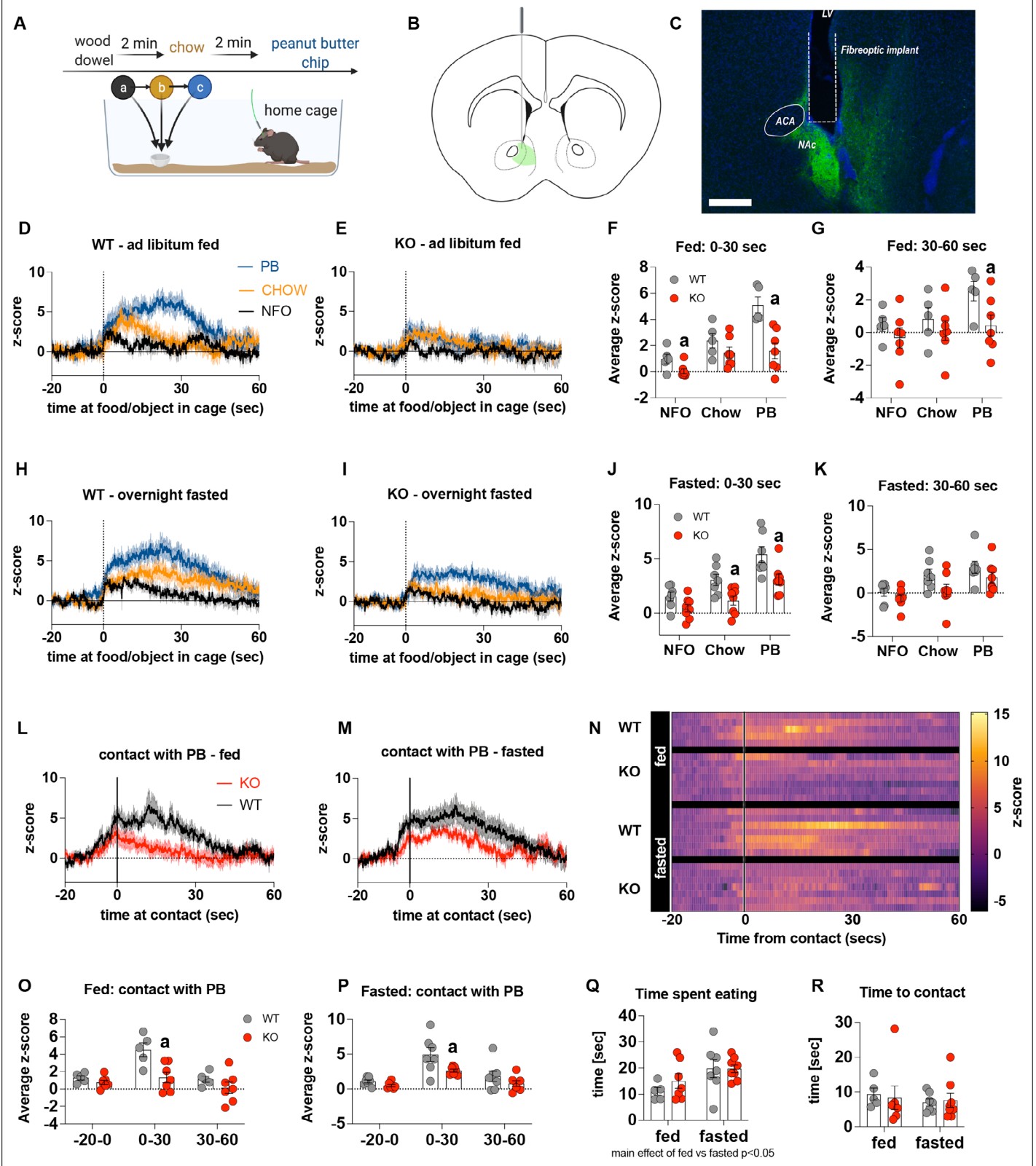

**Figure 2.** Impaired metabolic sensing in AgRP neurons affects dopamine release in the nucleus accumbens. Experimental design to examine dopamine signalling in nucleus accumbens (**A**). Female (3 WT, 4 KO) and male (4 WT, 4 KO) mice, trained to receive peanut butter chips, were tethered to a fiber optic cable in their home cage. Following acclimation, dopamine responses in the nucleus accumbens to a non-food object (NFO; wood dowel, a, black), standard chow (b, tangerine), and peanut butter chip (PB, c, blue) were measured. Schematic of GRAB-DA (AAV-hSyn-DA4.3) injection site

*Figure 2 continued on next page*

*Figure 2 continued*

in the nucleus accumbens (**B**) and fiber placement [scale bar = 500 µm] (**C**). Average z-scored dopamine release traces aligned (time = 0) to object dropping into cage of WT and KO fed mice (**D, E**) or WT and KO fasted (**H, I**) mice. The average z-score of WT and KO ad libitum-fed mice from 0 to 30 s (**F**) and 30–60 s (**G**) in response to a non-food object (NFO), standard chow, or peanut butter chip (PB) dropped into cage. The average z-score of WT and KO fasted mice from 0 to 30 s (**J**) and 30–60 s (**K**) in response to a non-food object (NFO), standard chow, or peanut butter chip (PB) dropped into cage. Average z-score dopamine release traces aligned to first contact with peanut butter chip (PB) from fed (**L**) and fasted (**M**) WT and KO mice, with individual traces shown in a heatmap (**N**). The average z-score of WT and KO ad libitum-fed mice from –20–0, 0–30, and 30–60 s (**O**), relative to first contact. The average z-score of WT and KO fasted mice from –20–0, 0–30, and 30–60 s (**P**), relative to first contact. Time spent eating during the recording session (**Q**) and time passed between drop and contact with peanut butter (**P**). Data± SEM, two-way ANOVA with Tukey's post hoc analysis (**F, G, J, K, O, P, Q** – main effect of fasting, p < 0.05; **R**); a – significant compared to WT, p < 0.05. Dashed lines at time = 0 (**D, E, H, I**) represent the time at which food or an object was dropped into the cage. A continuous line at time = 0 (**L, M**) represents the time at which mice contact PB chip. *Videos 3–6* accompany this figure.

The online version of this article includes the following source data for figure 2:

**Source data 1.** Impaired metabolic sensing in AgRP neurons affects dopamine release in the nucleus accumbens.

reflects the stage at which impaired metabolic sensing in AgRP neurons can no longer accurately report homeostatic state and energy need appropriately. Collectively, these results suggest that the appropriate detection of homeostatic state via metabolic sensing in AgRP neurons affects motivation for a palatable food reward.

To establish that deficits in motivation were associated with impaired dopamine release in the NAc, we used FED3 as it allows for programmable transitor-transitor logic (TTL) output to synchronise nose poking and pellet retrieval with measurements of dopamine release by GRAB-DA photometry (*Figure 4*). Dopamine release was measured during a progressive ratio allowing alignment of dopamine release to rewarded or non-rewarded nose pokes. In fed and fasted WT mice, a rewarded nose poke significantly increased NAc dopamine release, as assessed by average z-score 0–60 s after a poke event compared to 5 s prior to poke (*Figure 4D, H, F, G and J*). However, a rewarded nose poke in fed or fasted KO mice did not significantly increase NAc dopamine release within 60 s of the nose poke compared to 5 s prior to poke (*Figure 4E1, F, G and J*). In addition, NAc dopamine release was significantly reduced during the 60-s period after a rewarded nose poke in KO compared to WT mice in both fed and fasted states (*Figure 4F and G*). The time between nose pokes was lower in fasted mice compared to fed mice (main effect; *Figure 4P*), but no genotype differences were observed.

We also analysed dopamine release aligned to pellet retrieval (*Figure 4K, L, M, N, O, Q and R*) and observed genotype differences in fed (*Figure 4K and M*), and fasted (*Figure 4L and N*) mice. Dopamine release was significantly lower 0–10 s after pellet retrieval in KO mice compared to WT mice (*Figure 4K, L, M and N*), but not different before pellet retrieval suggesting that reduced dopamine in KO is due to sucrose pellet consumption itself. Interestingly, time to pellet retrieval was significantly longer in fed KO mice compared to fed WT mice (*Figure 4Q*). These results are consistent with progressive ratio data, in which breakpoint, active nose pokes and pellets collected are all significantly

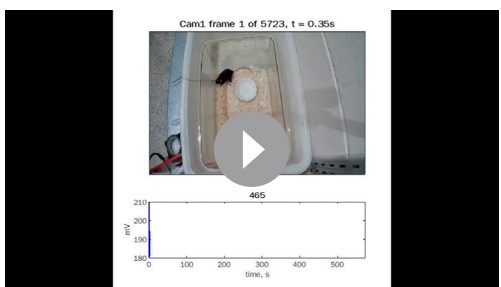

**Video 3.** GRAB-DA activity (dopamine release) time locked to behaviour at 10 x normal speed in an ad libitum-fed WT mouse. The mouse is first exposed to wood dowel, followed by chow, followed by a ~ 70 mg PB chip. The video shows raw data collected in mV at the photoreceiver prior to df/f calculations.

https://elifesciences.org/articles/72668/figures#video3

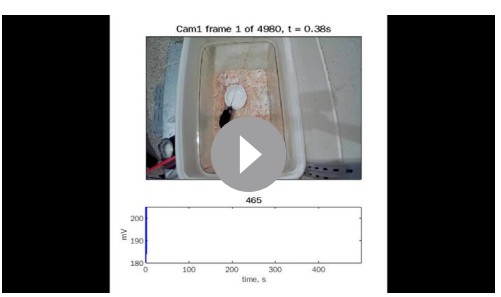

**Video 4.** GRAB-DA activity (dopamine release) time locked to behaviour at 10 x normal speed in a fasted WT mouse. The mouse is first exposed to wood dowel, followed by chow, followed by a ~ 70 mg PB chip. The video shows raw data collected in mV at the photoreceiver prior to df/f calculations.

https://elifesciences.org/articles/72668/figures#video4

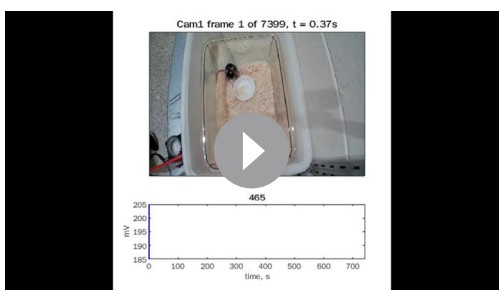

**Video 5.** GRAB-DA activity (dopamine release) time locked to behaviour at 10 x normal speed in an ad libitum-fed KO mouse. The mouse is first exposed to wood dowel, followed by chow, followed by a ~ 70 mg PB chip. The video shows raw data collected in mV at the photoreceiver prior to df/f calculations.
https://elifesciences.org/articles/72668/figures#video5

lower in KO mice compared to WT mice. Thus, reduced NAc dopamine release in response to pellet retrieval underpins reduced motivation for a palatable food reward in KO mice.

Previous studies have suggested that neural encoding of non-caloric and caloric solutions is differentially processed in the NAc and dorsal striatum respectively (*Tellez et al., 2016*). Next, we explored dopamine release in the dorsal striatum of WT and KO mice under fed and fasted conditions. We used the same protocol to measure acute dopamine release in the dorsal striatum in response to a non-food object (wood dowel), chow food and a PB chip with each presentation separated by 2 min (*Figure 5*). Although we observed a main effect for a PB chip to acutely increase dopamine release in the dorsal striatum 0–30 s and 30–60 s after food or object presentation, there were no significant genotype differences in either the fed or fasted state (*Figure 5D–K*). When data were aligned to contact with PB, we observed no genotype differences in contact with PB in fed or fasted mice at any time point (−20–0, 0–30, 30–60 s; *Figure 5L, M and S*). However, we observed a main effect of time to increase dopamine release at 0–30 s in fed and fasted mice (*Figure 5O and P*) independent from genotype showing that PB consumption elicits an acute increase in dopamine release in the dorsal striatum. Similar to data collected from the NAc experiments, no differences in time to contact with a PB chip or time spent eating a PB chip (5Q, R) were observed.

We also examined dopamine release in the dorsal striatum during a progressive ratio schedule (*Figure 6*). Aligning dorsal striatal dopamine to nose pokes showed a significant increase in dopamine release 0–10 seconds after a rewarded nose poke in both fed and fasted mice, independent from genotype (main effect of time; p < 0.05; *Figure 6D–I*). When aligned to pellet retrieval, dorsal striatal dopamine peaked immediately prior to retrieval (*Figure 6J–K*) with no significant genotype differences in average z-score prior to or after pellet retrieval (*Figure 6L and M*). Our data collected from the dorsal striatum suggest that impaired metabolic sensing in AgRP neurons does not affect acute dopamine release after food/object presentation (0–60 s) or during a PR (0–10 s). However, our photometry approach cannot assess longer term changes, which may be an important factor since a study using human subjects showed that there are two temporally and spatially distinct dopamine responses to ingesting a milkshake (*Thanarajah et al., 2019*). An immediate orosensory response occurs in response to taste and a second delayed post-ingestive response occurs in response to calories, which is localised to the dorsal striatum. To investigate this, we employed a positron electron tomography (PET) method using radiolabelled 18F-fDOPA (*Figure 6N*). First, we measured dynamic basal dopamine uptake in ventral (nucleus accumbens) and dorsal striatum in fasted mice without reward presentation and observed no differences in fDOPA uptake (*Figure 6—figure supplement 1*) suggesting impaired metabolic sensing in AgRP neurons does not affect baseline uptake parameters. However, in response to PB chip consumption, we detected an increase in fDOPA accumulation 30 min after starting in the dorsal striatum of WT but not in KO mice (*Figure 6P*) and no differences in the ventral striatum (*Figure 6O*). These studies suggest that impaired metabolic-sensing in AgRP neurons

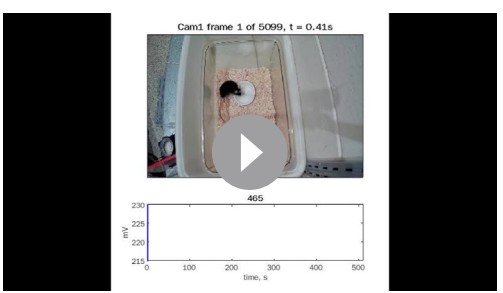

**Video 6.** GRAB-DA activity (dopamine release) time locked to behaviour at 10 x normal speed in a fasted KO mouse. The mouse is first exposed to wood dowel, followed by chow, followed by a ~ 70 mg PB chip. The video shows raw data collected in mV at the photoreceiver prior to df/f calculations.
https://elifesciences.org/articles/72668/figures#video6

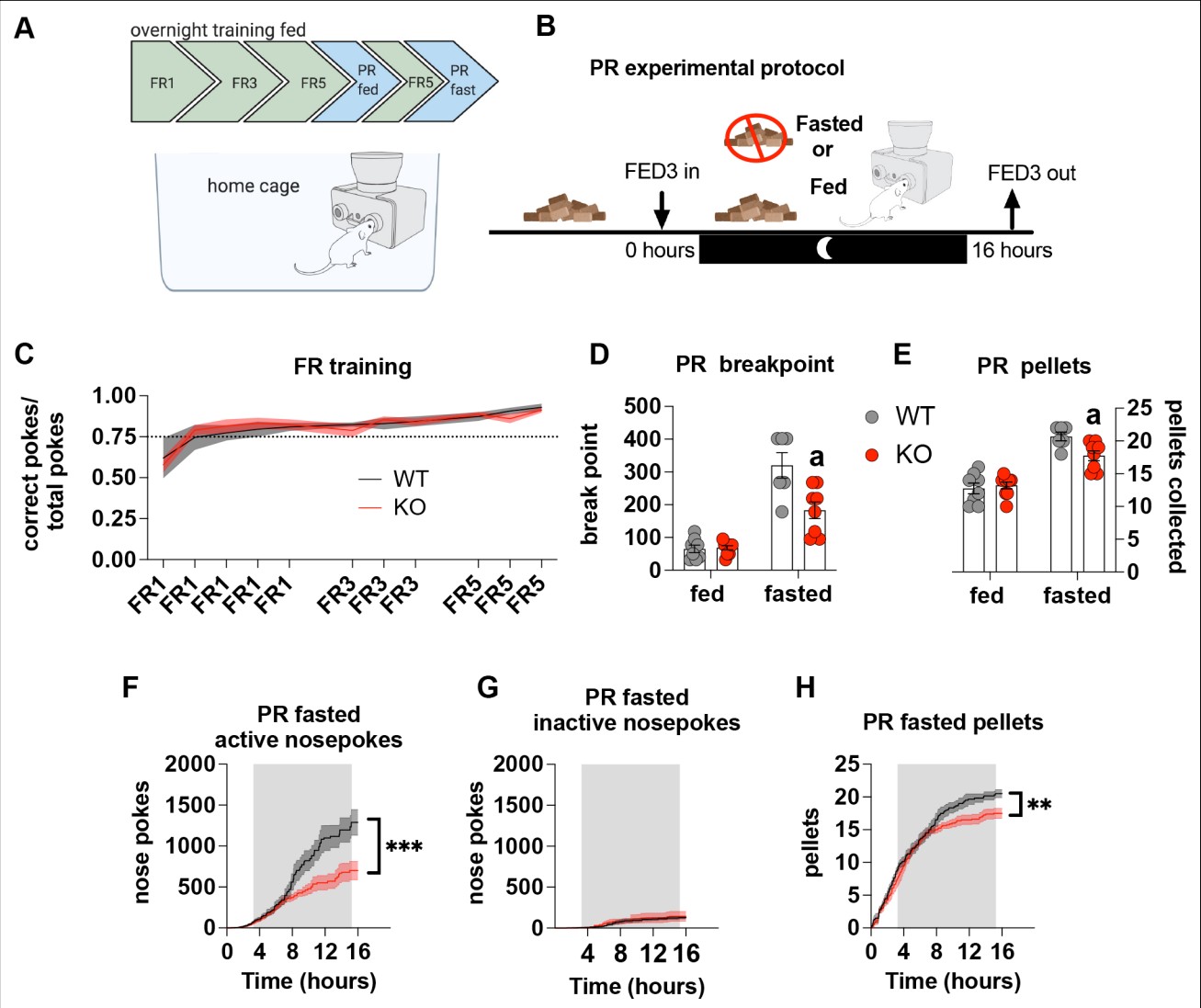

**Figure 3.** Impaired metabolic sensing in AgRP neurons affects motivation for sucrose rewards during fasting. Mice (8 WT, 9 KO) were trained to nose poke for sucrose rewards using fixed ratio schedule (**A** - FR1, FR3, FR5) to reliably nose poke on average 75% correct for three consecutive nights (**C**), before undergoing PR schedules with or without chow accessible. For PR under fasting conditions, mice were not fasted prior to testing but progressively fasted from the beginning until the end of the PR session (**B**). This important distinction tests the motivation to the progressively increasing energy deficit rather than in response to an already higher energy deficit. Breakpoint from fed and fasted mice (**D**) and pellet retrieval at the end of PR sessions (**E**). The number of active nose pokes (**F**), inactive (**G**) and pellets collected during PR fasted sessions over time. Data± SEM, two-way repeated measures ANOVA with Tukey's post hoc analysis (**F, H** – main effect of genotype). Data± SEM, two-way ANOVA with Tukey's post hoc analysis (**D, E**). a – significant compared to WT, p < 0.05, ** p < 0.01, *** p < 0.001.

The online version of this article includes the following source data for figure 3:

**Source data 1.** Impaired metabolic sensing in AgRP neurons affects motivation for sucrose rewards during fasting.

restricts a post-ingestive dopamine response, ~ 30–35 min after consumption of a PB chip consumption, in the dorsal striatum.

## Discussion

In this study, we show that AgRP neurons require appropriate metabolic sensing of available calories to engage midbrain dopamine circuits and increase food motivation. We demonstrate that *Crat* deletion in AgRP neurons is a valid model of abnormal metabolic-sensing since it impairs glucose-sensing using ex vivo electrophysiological recordings and impairs in vivo AgRP GCaMP6 responses

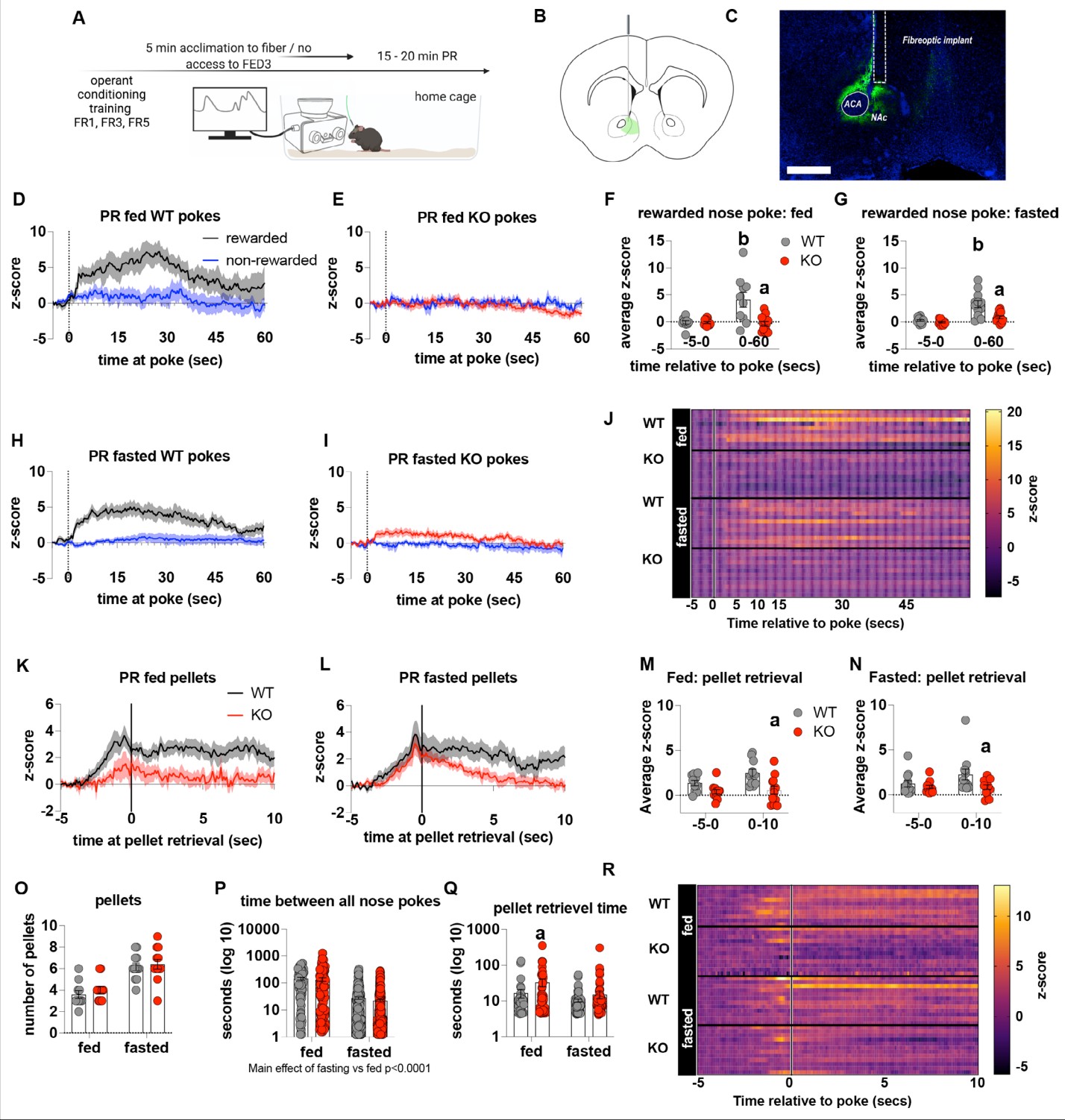

**Figure 4.** Impaired metabolic sensing in AgRP neurons reduces dopamine release in the nucleus accumbens during a progressive ratio session. Experimental design (**A**) - mice with fiberoptic implant in the nucleus accumbens were trained to nose poke for a sucrose pellet prior to experimental testing. During testing, mice were tethered to a fiberoptic cable in their home cage and after 5 min acclimation, mice gained access to FED3 on PR schedule. Schematic of GRAB-DA (AAV-hSyn-DA4.3) injection site in the nucleus accumbens (**B**) and fiber placement [scale bar = 500 μm] (**C**). Combined average dopamine release response aligned (time = 0) to rewarded and non-rewarded correct nose pokes during a PR session from WT (**D**; n = 10 mice) and KO (**E**; n = 12 mice) ad libitum-fed mice and WT (**H**; n = 12 mice) or KO (**I**, n = 12 mice) after an overnight fast. The average z-score of rewarded nose pokes from WT and KO ad libitum-fed mice at –5–0 s and 0–60 s (**F**) or from WT and KO fasted mice at –5–0 s and 0–60 s (**G**). Heat maps of

*Figure 4 continued on next page*

*Figure 4 continued*

averaged responses from each experimental animal aligned to nose poke (**J**). Average z-score dopamine release traces aligned to pellet retrieval (time = 0) from ad libitum-fed (**K**) or fasted (**L**) WT and KO mice. The average z-score at –5–0 s and 0–10 s after pellet retrieval in ad libitum-fed (**M**) or fasted mice (**N**). Rewarded nose pokes /Pellets received (**O**) and the time between rewarded and non-rewarded nose pokes during the recording PR session in ad libitum-fed or fasted WT and KO mice (**P**). WT and KO mice with pellet retrieval time for fed and fasted WT and KO mice shown in Q. Heatmaps of averaged responses from each experimental animal aligned to pellet retrieval (**R**). Data± SEM, two-way ANOVA with Tukey's post hoc analysis. a – significant compared to WT, $p < 0.05$, b – significant compared to WT –5–0 s, $p < 0.05$. Dashed lines at time = 0 (**D, E, H, I**) represent the time at which a nose poke was made. A continuous solid line at time = 0 (**L, M**) represents the time at which mice collected a pellet from the pellet dispenser.

The online version of this article includes the following source data for figure 4:

**Source data 1.** Impaired metabolic sensing in AgRP neurons reduces dopamine release in the nucleus accumbens during a progressive ratio session.

to IP glucose injection and repeated palatable food intake. An important role for *Crat* in metabolic sensing is supported by studies deleting *Crat* from myocytes, as this diminishes the switch from fatty acid metabolism to glucose metabolism after pyruvate administration (*Muoio et al., 2012*).

There has been a shift in the understanding of AgRP neuronal function since it was discovered that the integration of sensory information suppresses AgRP neuronal activity prior to food consumption (*Betley et al., 2015*; *Mandelblat-Cerf et al., 2015*; *Chen et al., 2015*). Intriguingly, in our studies, the first presentation of PB to WT and KO mice produces a similar reduction in AgRP neural activity suggesting normal integration of sensory and taste information, in the absence of post-ingestive interoceptive information. The subsequent fall in AgRP activity from WT mice after initial PB consumption shows that WT mice integrate additional interoceptive information based on post-ingestive feedback from PB consumption. However, KO mice do not show any greater suppression of AgRP neural activity with repeated PB exposure and consumption. Our results are supported by several studies illustrating that post-ingestive feedback from gut-derived signals is required to maintain the suppression of AgRP activity in response to consumption (*Su et al., 2017*; *Beutler et al., 2017*; *Goldstein et al., 2021*). Su et al demonstrated that the consumption of caloric gels, but not non caloric gels, is required for the sustained suppression of AgRP neuron activity, the magnitude of which is proportional to the calories obtained. In these studies the caloric value of a gel was learnt in a single trial (*Su et al., 2017*), similar to the response seen in WT mice after first exposure to PB in our studies. It should be noted, however, that we cannot rule out the possibility of reduced synaptic inhibitory input on to AgRP neurons in fasted KO mice after initial PB presentation, although no differences in baseline inhibitory or excitatory input on to AgRP neurons were observed in WT and KO mice. The majority of inhibitory input to AgRP neurons comes from GABAergic leptin-receptor expressing DMH (DMH[LEPR]) neurons (*Garfield et al., 2016*). Importantly, DMH[LEPR] neurons provide inhibitory input to AgRP neurons in response to sensory cues but with only a minimal response to metabolic signals (*Berrios et al., 2021*), arguing against the possibility of altered inhibitory input in AgRP neurons from KO mice in our study. It is also possible that *Crat* deletion from AgRP neurons alters the post-synaptic response to GABAergic inhibitory inputs since altered mitochondrial glucose metabolism, as observed after crat deletion, affects GABA metabolism (*Cavalcanti-de-Albuquerque et al., 2021*).

Similar to recently published observations (*Goldstein et al., 2021*), IP glucose injection suppressed AgRP neural activity in WT and KO mice, however activity quickly returned to pre-injection levels in WT fasted but not KO fasted mice. Given that the integration of calorie information is required to sustain suppressed AgRP neural activity (*Chen et al., 2015*; *Su et al., 2017*; *Beutler et al., 2017*), our data highlight that calorie availability from IP glucose administration was insufficient to sustain the suppression of AgRP neural activity in fasted WT mice and activity returned to pre-glucose levels to encourage further food seeking. Although fasted KO mice reduced activity in response to IP glucose injection, AgRP activity did not return to baseline levels indicating an inability to sense further calorie need. Thus, we have generated a valid model of impaired metabolic sensing in AgRP neurons in vivo.

By using GRAB-DA photometry to measure dopamine release dynamics (*Sun et al., 2018*), we observed that KO had significantly attenuated acute dopamine release in the NAc in fed or fasted state, despite equal time spent consuming a PB chip. By contrast, we did not see any difference in acute dorsal striatum dopamine release between WT and KO mice. Thus, metabolic sensing in AgRP neurons is required to potentiate acute dopamine release in the NAc, extending previous observations that artificial activation of AgRP neurons increases dopamine release in response to food (*Alhadeff et al., 2019*).

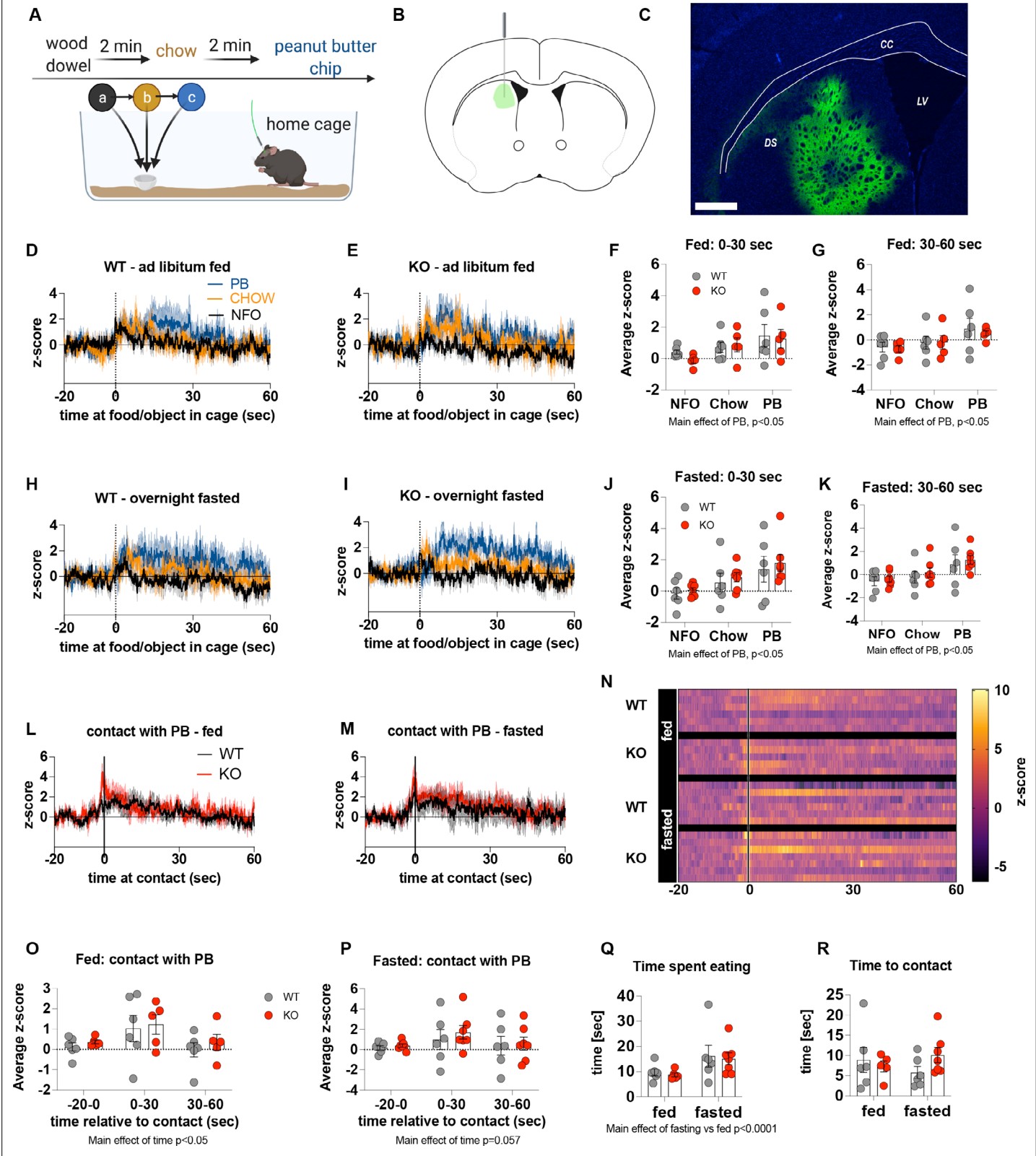

**Figure 5.** The effect of impaired metabolic sensing in AgRP neurons on dopamine release in the dorsal striatum. Experimental design to examine dopamine signalling in dorsal striatum (**A**): Female (2 WT, 3 KO) and male (4 WT, 4 KO) mice, trained to receive PB chips, were tethered to a fiber optic cable in their home cage. Following acclimation, dopamine responses in the dorsal striatum to a non-food object (NFO; wood dowel, a, black), standard chow (b, tangerine), and peanut butter chip (PB, c, blue) were measured. Schematic of GRAB-DA (AAV-hSyn-DA4.3) injection site in the dorsal striatum

*Figure 5 continued on next page*

*Figure 5 continued*

(**B**) and fiber placement [scale bar = 500 µm] (**C**). Average z-scored dopamine release traces aligned (time = 0) to object dropping into cage of WT and KO fed mice (**D, E**) or WT and KO fasted (**H, I**) mice. The average z-score of WT and KO ad libitum-fed mice from 0 to 30 s (**F**) and 30–60 s (**G**) in response to a non-food object (NFO), standard chow, or peanut butter chip (PB) dropped into cage (n = 6 WT, n = 5 KO). The average z-score of WT and KO fasted mice from 0 to 30 s (**J**) and 30–60 s (**K**) in response to a non-food object (NFO), standard chow, or peanut butter chip (PB) dropped into cage (n = 6 WT, n = 7 KO). Average z-score dopamine release traces aligned to first contact with peanut butter chip (PB) from fed (**L**) and fasted (**M**) WT and KO mice, with individual traces shown in a heatmap (**N**). The average z-score of WT and KO ad libitum-fed (**O**) and fasted mice from –20–0, 0–30, and 30–60 s (**P**), relative to first contact. Time spent eating during the recording session (**Q**) and time passed between drop and contact with peanut butter (**R**). Dashed lines at time = 0 (**D, E, H, I**) represent the time at which food or an object was dropped into the cage. A continuous line at time = 0 (**L, M**) represents the time at which mice contact PB chip. Data± SEM, two-way ANOVA with Tukey's post hoc analysis (F, G, J, K – main effects of PB, p < 0.05; (**O**) – main effect of time, p < 0.05; (**P**) – main effect of time p = 0.057; (**Q**) – main effect of fasting vs fed p < 0.05).

The online version of this article includes the following source data for figure 5:

**Source data 1.** The effect of impaired metabolic sensing in AgRP neurons on dopamine release in the dorsal striatum.

Importantly, impaired metabolic sensing in AgRP neurons also attenuated progressive ratio operant responding for sucrose rewards during fasting with KO mice showing reduced breakpoint achieved, active nose pokes and pellets consumed. Although the motivation to obtain food is associated with chemo- or opto-genetic activation of AgRP neurons (*Krashes et al., 2011*; *Chen et al., 2016*), our studies show metabolic sensing of homeostatic state is required for translating low energy availability into a dopamine-driven motivational action to consume sucrose.

Using FED3 coupled with GRAB-DA photometry, we measured dopamine release in the NAc or dorsal striatum during a progressive ratio session in response to rewarded and non-rewarded nose pokes. In the NAc, dopamine release increased prior to pellet retrieval consistent with FED3-motor activity acting as an auditory cue for pellet delivery, given the well-known role of cue-induced dopamine release in the NAc (*Schiffer et al., 2009*; *Covey and Cheer, 2019*). Intriguingly, cue-evoked dopamine release in the NAc reflects expected dopamine-mediated reinforcement rather than the actual magnitude of dopamine neuronal activation (*Covey and Cheer, 2019*). There were no differences in average z-score prior to pellet retrieval, however average z-score was reduced after pellet retrieval in KO mice under fasted conditions. These observations show that metabolic sensing in AgRP neurons does not influence the response to the reward predicting cue, but to the reward itself, which is driven by taste and caloric value, as previously described (*Chen et al., 2015*; *Goldstein et al., 2021*; *Fu et al., 2019*). The combination of reduced breakpoint responding and lower dopamine release in the NAc to pellet retrieval is consistent with the known role of NAc dopamine release to drive food seeking (*Roitman et al., 2004*) and to assign an appropriate investment of effort to the available reward (*Hamid et al., 2016*; *Howe et al., 2013*; *Mohebi et al., 2019*). In the dorsal striatum, dopamine ramping was observed prior to pellet retrieval without any genotype differences, which dropped to baseline soon after pellet retrieval consistent with previous reports (*London et al., 2018*). One potential limitation from our studies is that reduced dopamine release in KO mice may reflect less active AgRP neurons at baseline or during fasting. Although our ex vivo electrophysiology studies show no differences at baseline, they cannot rule out in vivo differences that could not be addressed with ex vivo slice recording or population calcium changes using GCaMP6.

With photometry, we were unable to detect any differences in dopamine release in the dorsal striatum yet studies in humans show palatable food elicits an immediate orosensory dopamine response and a delayed dorsal striatum dopamine response (*Thanarajah et al., 2019*). Our photometry approach only examined a 60-s window after PB consumption or a 10-s window after pellet retrieval. Indeed, after these short time periods dopamine release had already returned to baseline, making it impossible to utilise photometry to measure long-term changes in dopamine function. To address longer term changes in dopamine function we used 18 F fDOPA in PET/CT studies and revealed reduced dopamine uptake 30 min after PB consumption in fasted KO mice compared to WT mice, with no difference in the ventral striatum. Thus, we have identified an interesting separation in the temporal relationship of metabolic sensing in AgRP neurons and dopamine release in the NAc or dorsal striatum. Impaired metabolic sensing in AgRP neurons causes an acute attenuation of dopamine release in the NAc but not the dorsal striatum whereas it causes a longer term attenuation in the dorsal striatum but not the ventral striatum, which encompasses the NAc. The exact mechanisms for these differences are unknown, although it is encouraging to see a similar temporal separation in an

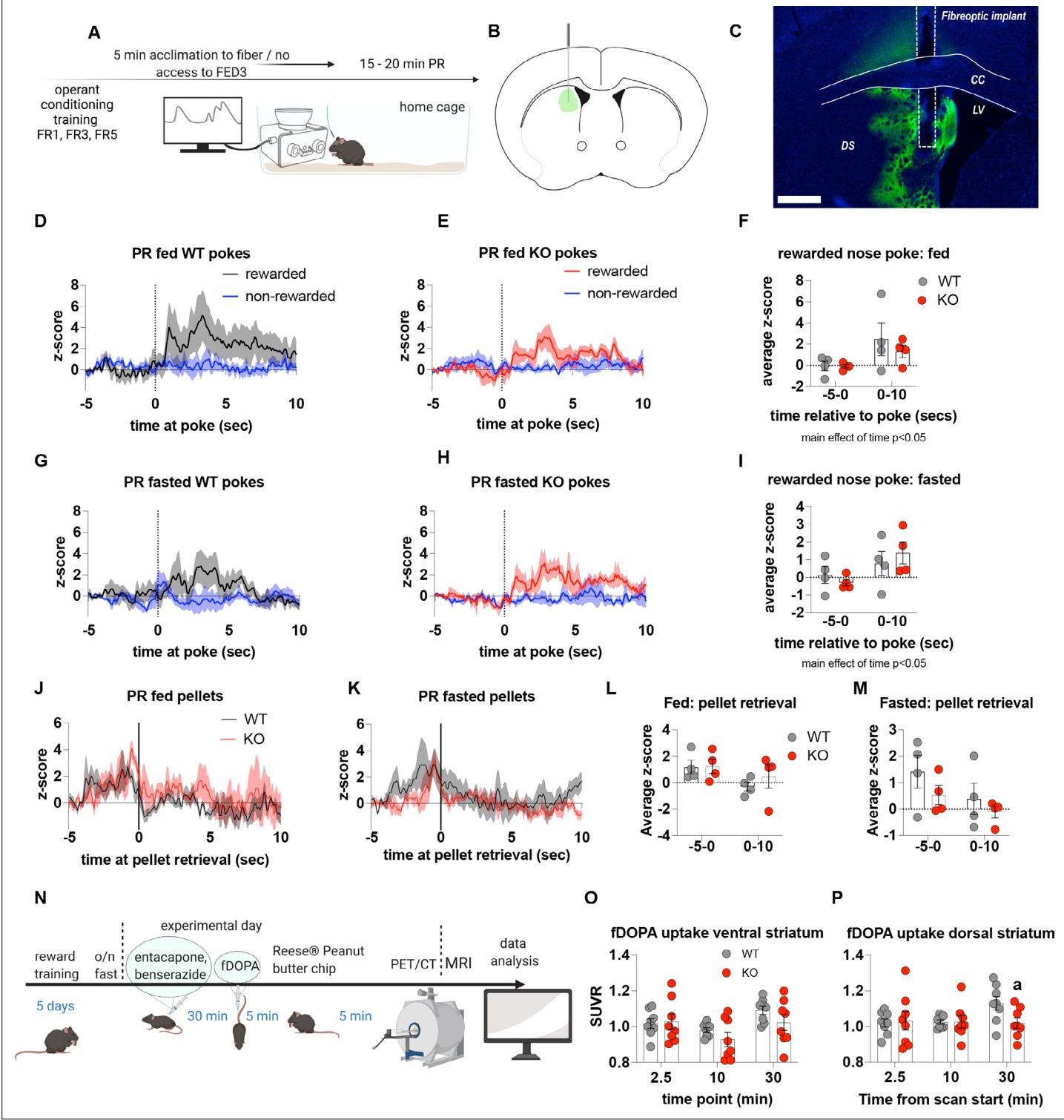

**Figure 6.** Impaired metabolic sensing in AgRP neurons does not affect dopamine release in the dorsal striatum during a progressive ratio session. Experimental design (**A**) - mice with a fiberoptic implant in the dorsal striatum were trained to nose poke for a sucrose pellet prior to experimental testing. During testing, mice were tethered to a fiber optic cable in their home cage and after 5 min acclimation, mice gained access to FED3 on PR schedule. Schematic of GRAB-DA (AAV-hSyn-DA4.3) injection site in the dorsal striatum (**B**) and fiber placement [scale bar = 500 μm] (**C**). After 5 min acclimation mice gained access to FED3 on PR schedule. Combined average dopamine response of rewarded and non-rewarded correct nose pokes of 4 WT (**D**) and 4 KO (**E**) in fed state and 4 WT (**G**) and 4 KO (**H**) after overnight fast. The average z-score of rewarded nose pokes from WT and ad libitum-fed mice at –5–0 s and 0–10 s (**F**) or from WT and fasted mice at –5–0 s and 0–10 s (**I**). Average z-score dopamine release traces aligned to pellet retrieval

*Figure 6 continued on next page*

*Figure 6 continued*

(time = 0) from ad libitum-fed (**J**) or fasted (**K**) WT and KO mice. The average z-score at –5–0 s and 0–10 s relative to pellet retrieval in ad libitum-fed (**L**) or fasted (**M**) WT and KO mice. Experimental design for fDOPA PET scan (**N**): Mice (WT n = 8 male, KO n = 8 male) were trained to receive Reese peanut butter (PB) chips and then fasted overnight before the experimental day. On the experimental day, mice received ip injection of benserazide and entacapone to prevent peripheral breakdown of fDOPA (iv injection 30 min later). Mice were allowed to recover 5 min and then given one peanut butter chip, which they ate within 5 min. Then mice were anaesthetised and prepared for PET/CT scan and dopamine uptake in dorsal and ventral striatum measured. Baseline fDOPA uptake in ventral striatum (**O**) and dorsal striatum (**P**). Dashed lines at time = 0 (**D, E, G, H**) represent the time at which a nose poke was made. A continuous line at time = 0 (**J, K**) represents the time at which mice collected the pellet from the pellet dispenser. Data± SEM, two-way ANOVA with Tukey's post hoc analysis (**F, I, L, M** – main effect of time, $p < 0.05$). a – significant compared to WT, $p < 0.05$.

The online version of this article includes the following source data and figure supplement(s) for figure 6:

**Source data 1.** Impaired metabolic sensing in AgRP neurons does not affect dopamine release in the dorsal striatum during a progressive ratio session.

**Figure supplement 1.** Dynamic basal dopamine uptake in dorsal (**A**) and ventral striatum (**B**) in fasted mice without reward presentation and observed no differences in fDOPA uptake.

**Figure supplement 1—source data 1.** Dynamic basal dopamine uptake in dorsal and ventral striatum in fasted mice without reward presentation and observed no differences in fDOPA uptake.

orosensory dopamine response and a delayed dorsal striatum dopamine response in humans (*Thanarajah et al., 2019*). It is possible that the dorsal striatum response is delayed due to a requirement for metabolic processing in the gut or peripheral tissues. For example, disrupting peripheral glucose oxidation suppresses dorsal striatum dopamine efflux during sugar intake (*Tellez et al., 2013*) and we have previously observed impaired glucose oxidation in response to refeeding after fasting in mice lacking *Crat* in AgRP neurons (*Reichenbach et al., 2018c*; *Reichenbach et al., 2018b*; *Reichenbach et al., 2018a*). Moreover, AgRP neurons respond to sensory cues within seconds yet require a slightly longer time frame (~minutes) to respond effectively to post-ingestive signals from the gut, such as nutrients and hormones released into the blood or via neural pathways (*Su et al., 2017*; *Beutler et al., 2017*; *Goldstein et al., 2021*; *Bai et al., 2019*). Thus, we predict that impaired metabolic sensing in AgRP neurons affects both the immediate acute response to reward consumption, via dopamine signalling in the NAc, as well as a slower response, which involves the integration of post-ingestive signals of calorie content in regions such as the dorsal striatum. In support of this, dopamine release in the dorsal striatum depends on calorie content, independent from taste, and is modulated by direct calorie infusion into the gut and gut-derived vagal afferents (*Tellez et al., 2016*; *de Araujo et al., 2008*; *Han et al., 2016*; *Han et al., 2018*).

Furthermore, we predict that both short term NAc dopamine release and longer-term dorsal striatum dopamine release may be required to manifest a behavioural change in motivation, both of which require metabolic sensing in AgRP neurons. This comes from our observations that NAc dopamine release to a rewarded nose poke is lower in both fed and fasted KO mice, however reduced motivation for sucrose pellets in the PR session only manifests 6 hr into the fasted period. Indeed, we only observed changes in fDOPA accumulation in the dorsal striatum in mice fasted overnight.

Although artificial activation of AgRP neurons increases VTA dopamine activity (*Mazzone et al., 2020*), an unanswered question remains, as to how AgRP neurons influence mesolimbic dopamine signalling. Given that very few, if any, AgRP terminals are found in the adult VTA (*Dietrich et al., 2012*) and the slow nature of AgRP activity at the MC4R (*Krashes et al., 2013*), it seems more likely that the effects of AgRP neurons on mesolimbic dopamine activity is driven through either NPY or GABA release at downstream targets. NPY release is required for the ongoing ability of AgRP neurons to drive positive reinforcement and hunger signalling after the suppression of AgRP neuronal firing (*Chen et al., 2016*; *Chen et al., 2019*), suggesting it is the prime candidate to modulate dopamine release in NAc. In summary, we show that metabolic sensing in AgRP neurons is required to transmit interoceptive metabolic information into dopamine release in the NAc and dorsal striatum, albeit over different time frames, and to increase motivated behaviour for sucrose rewards.

## Materials and methods
### Animals

All experiments were conducted in compliance with the Monash University Animal Ethics Committee guidelines. Male and female mice were kept under standard laboratory conditions with free access to

food (chow diet, catalog no. 8720610, Barastoc Stockfeeds, Victoria, Australia) and water at 23°C in a 12 hr light/ dark cycle and were group-housed to prevent isolation stress unless otherwise stated. No sex differences were observed and data from both male and female mice were grouped together, as indicated in figure legends. All mice were aged 8 weeks or older for experiments unless otherwise stated. *Agrp-ires-cre* mice were obtained from Jackson Laboratory *Agrp^tm1(cre)Low/J* (stock no. 012899) and bred with NPY GFP mice (B6.FVB-Tg(Npy-hrGFP)1Lowl/J; stock number 006417; The Jackson Laboratory, Maine, USA). AgRP-ires-cre::NPY GFP mice were then crossed with *Crat^fl/fl* mice donated by Randall Mynatt (Pennington Biomedical Research Center, LA, USA) in order to delete *Crat* from AgRP neurons (*Agrp^cre/wt^::Crat^fl/fl* mice; designated as KO). *Agrp^wt/wt^::Crat^fl/fl* littermate mice were used as control animals (designated as WT). For in vivo photometry *Agrp^cre/wt^::Crat^wt/wt* mice; designated as WT to allow for cre-dependent expression of GCaMP6s specifically in AgRP neurons (KO same as above). No statistical methods were used to determine samples sizes. Experimental mice for operant behaviour and photometry were individual housed.

## Electrophysiology

Animals were fasted overnight and 250-μm-thick coronal hypothalamic brain slices containing the ARC were prepared from seven male AgRP *Crat* KO and 7 WT mice (8–12 weeks) expressing GFP in NPY neurons, and stored at room temperature before transferral to the recording chamber. Slices were continuously superfused at 4–5 ml/min with oxygenated (95% $O_2$, 5% $CO_2$) artificial cerebrospinal fluid (aCSF) of the following composition (in mM): NaCl 127, KCl 1.9, $KH_2PO_4$ 1.2, $CaCl_2$ 2.4, $MgCl_2$ 1.3, $NaHCO_3$ 26, D-Glucose 2, Mannitol 8 (310 mOsm, pH 7.4). Hypothalamic neurons were visualised under IR illumination using a 63 x or 40 x water immersion objective on an upright microscope (Axioskop 2, Zeiss) and an Axiocam MRm camera (Zeiss). AgRP neurons were identified using a GFP filter set. Patch pipettes (8–11 MΩ) were pulled from borosilicate glass capillaries (Harvard Apparatus) and filled with intracellular solution containing (in mM): K-gluconate 140, KCl 10, EGTA 1, HEPES 10, Na-ATP 2, Na-GTP 0.3 (300 mOsm and pH 7.3, adjusted with KOH). Whole-cell current clamp recordings were made using the MultiClamp 700B amplifier, digitised with the Digidata 1,550B interface, and acquired in pClamp 10.6 at 5 kHz sampling rate (Axon Instruments). To test the influence of an elevated extracellular glucose concentration on neural activity, aCSF was prepared as described above with the following changes: D-Glucose 5 mM, Mannitol 5 mM, and bath-applied for 10–15 min. Data were analysed in Clampfit 10.6 (Axon Instruments) and plotted in Graphpad Prism 8.3. Figures were further prepared in Adobe Illustrator CC 2020.

## Fiber photometry

Mice for fiber photometry experiments were anaesthetised (2–3% isoflurane) and injected with Metacam (5 mg/kg) prior to placing into stereotaxic frame (Stoelting/Kopf) on heatpad (37°C) and viral injections were performed as previously described (*Mani et al., 2017*). Cre-dependent GCaMP6s (Addgene #100,845 AAV9-hSyn-FLEX-GCaMP6s-WPRE-SV40) was unilaterally injected into the ARC (–1.5 mm Bregma;±0.2 mm lateral; –5.6 mm ventral from surface of brain). Non-cre dependent dopamine sensor (GRAB-DA, YL10012-AAV9: AAV-hSyn-DA4.3) (*Sun et al., 2018*) was unilaterally injected in dorsal striatum (bregma 0.5 mm, midline 1.3 mm. skull –3.4 mm) or NAc (bregma 1.2 mm, midline 0.5 mm. skull –4.8 mm). Injections were 150 nl/side @25 nl/min, 5 min rest and ferrule capped fibers (400 μm core, NA 0.48 Doric, MF1.25 400/430–0.48) implanted above injection site and secured with dental cement (GBond, Japan). Mice had 2 weeks recovery before commencement of experiments.

All fiber photometric recordings were performed with a rig using optical components from Doric lenses controlled by Tucker Davis Technologies fiber photometry processor RZ5P. TDT Synapse software was used for data acquisition. Prior to experiments, baseline GCaMP6s or dopamine signal was measured and LED power was adjusted for each mouse to achieve approximately 200 mV for 465 nm (530 Hz) and 100 mV for 405 nm (210 Hz). This approach is designed to minimise the impact of variable GCaMP6s expression, tissue irradiance, surgery quality and distance of the fibre optic from GCaMP6s fluorescence. 405 nm was used as an isosbestic control, which is wavelength at which GCaMP6 excitation is independent from intracellular [$Ca^{2+}$], allowing for an assessment of motion artefact. For data analysis, we used a modified python code provided by TDT (*Lerner et al., 2015*; *Barker et al., 2017*). The modified code is available at GitHub(*Reichenbach, 2021*; copy archived at swh:1:rev:089ab86fb0586563bbd9cdaf0c6bc73b96ce905f). Briefly,

raw traces were down sampled and df/f (f465 nm-f405 nm/f405 nm) was calculated to detrend signal bleaching and remove any motion artefact. Individual z-scores around each precisely timed transistor-transistor logic (TTL)-triggered event were extracted to allow a standardised comparison between events. Z-score normalisation indicates the number of standard deviations a particular data point is away from a defined baseline mean and was calculated according to the following equation ($z = [x - x^\wedge]/S$); where the raw data point $x$ is subtracted from the mean of the baseline period $x$ divided by the standard deviation $S$ of the baseline period. The baseline period was defined as the period prior to a defined TTL-triggered behavioural event as described in results and figure legends.

To measure dopamine release to non-food-object/chow/peanut butter chip, single-housed mice with fiber implants were habituated to receiving Reese's peanut butter chips in the home cage (Macronutrient composition – Fat 29%, Carbohydrate 52%, Protein 3%). On the test day mice were connected to fiber photometry setup in their home cage and a small ceramic bowl placed inside. Recording started after 5 minutes acclimation period. In 2 min intervals, a small wood dowel (novel non-food object), a chow pellet, and a peanut butter chip were dropped into the ceramic bowel in that order. Mice were fed or fasted in random cross over design. Peanut butter chips were measured to ~70 mg per pellet and one pellet was given per trial. Mice consumed all of this peanut butter chip during each trial such that no differences in consumption were observed. Recordings were conducted during the light phase, with the majority towards the end of the light phase. GRAB-DA responses in WT and KO mice were similar on approach to PB and prior to pellet retrieval in both the NAc and dorsal striatum showing that genotype differences in response to palatable food or sucrose pellets could not be due to differences in GRAB-DA expression in the NAc. Moreover, a post mortem analysis was conducted to identify the localisation of GFP expression (*Figure 7*).

For fiber photometry using GCaMP6s in AgRP neurons, successful targeting of AgRP neurons was tested 2 weeks after surgery by injecting ghrelin (1 mg/kg). Baseline GCaMP6s signals were recorded 15 min prior to ip injection and mice presented with standard chow 90 min after injection. To measure AgRP responsiveness to palatable food, mice were presented with Reese's peanut butter chips. Mice were habituated to peanut butter chips for 5 days (1 × 70 mg pellet/day) unless otherwise stated as naïve (*Figure 1G–H* only). Multiple trails of this experiment were repeated over multiple days in fed and fasted mice (overnight fasting ~16 hr). To measure glucose responsiveness of AgRP neurons, fed or fasted mice were injected with glucose (2.25 g/kg) and AgRP neuronal activity recorded for 10 min.

## Operant conditioning task

For operant conditioning experiments, Feeding Experiment Devices version 3 (FED3)(*Matikainen-Ankney et al., 2021*) were placed overnight (16 hr) inside home cages under ad libitum conditions trained to reliably nose poke on fixed ratio (FR)1, FR3, and FR5 schedules (criteria to move to higher schedule was three consecutive days over 75% correct nose pokes) The dispensing of a sugar pellet (20 mg, 65% sucrose, 5TUT Test Diets, Saint Louis, Missouri, USA) was paired with an LED light cue. During these FR sessions, a nose poke in the 'active' hole resulted in the delivery of a sugar pellet and was paired with a LED light cue whereas a nose poke in the 'inactive' hold resulted in no programmed response. Importantly, mice were never food restricted during training to prevent this confounding the interpretation of our results. Once stable operant responding was established ( > 75% correct nose pokes on FR5) mice were placed on a progressive ratio (PR) schedule under ad libitum-fed conditions for a single overnight PR session (16 hr). PR sessions were based on a Richardson Roberts schedule (*Richardson and Roberts, 1996*) where the number of pokes required to obtain a reward increased after every completed trial in the following pattern; 1, 2, 4, 6, 9, 12, 15, 20, 25, 32, etc. After this single PR session, stable FR5 responding was re-established ( > 75% correct nose pokes, typically 1 FR session) and a second PR session was performed except this time mice were fasted 16 hr during to the PR session. Separate cohorts with fiber implants in the dorsal striatum or NAc were trained to receive sugar rewards as described above and dopamine responses to both nose pokes and sugar pellet retrieval were recorded in fed state or after an overnight fast. For GRAB-DA photometry during PR session, data are presented as the averaged dopamine response for each animal. On average mice collected ~3.5 pellets during the PR in ad libitum-fed conditions and ~6 pellets when fasted.

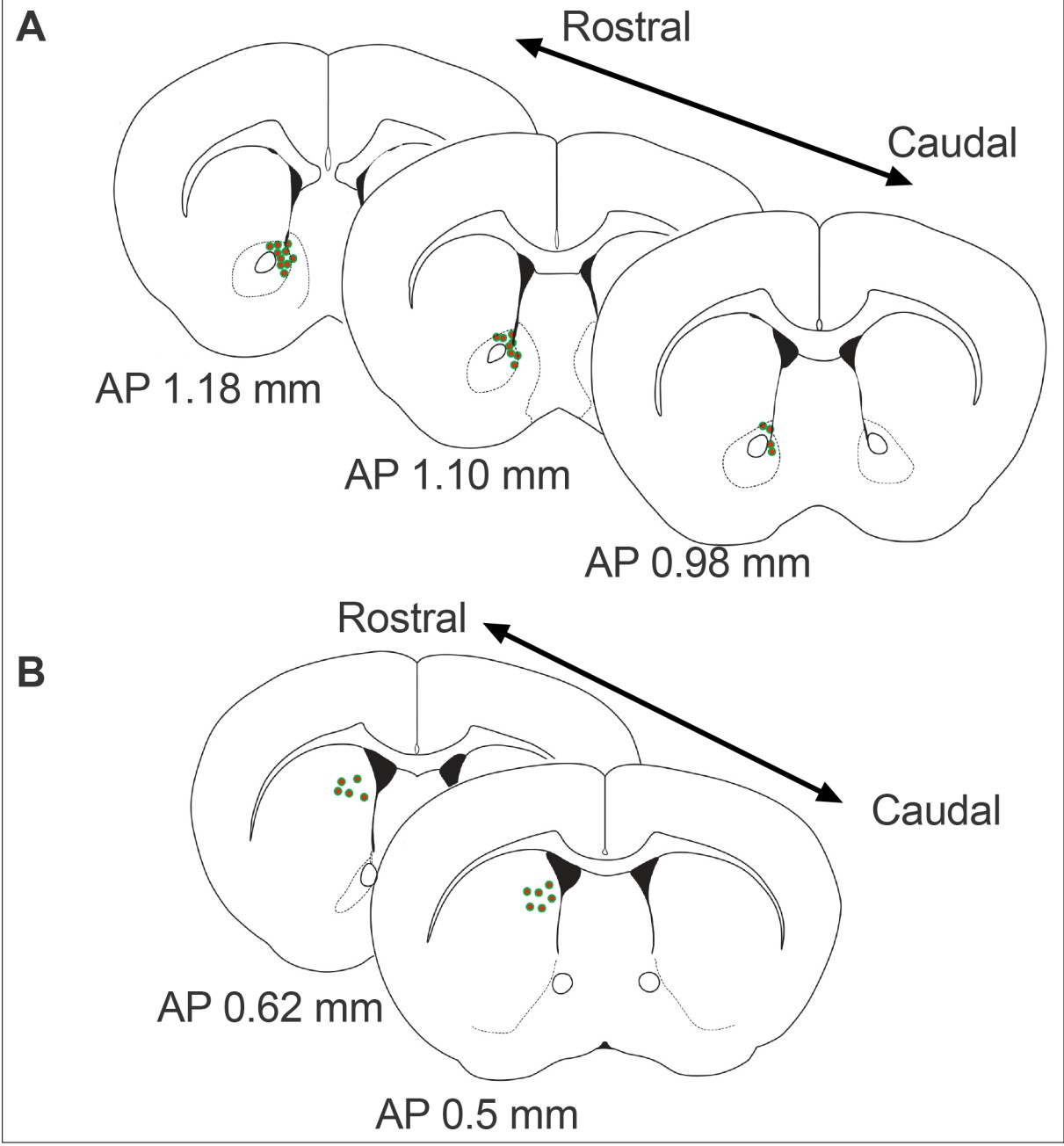

**Figure 7.** Schematic representation of the approximate viral injections sites for GRAB-DA in the Nucleus Accumbens (**A**) and dorsal striatum (**B**). AP = anterior posterior position relative to bregma.

### PET/CT and MRI

Mice were single housed and trained to receive a single Reese's peanut butter chip 6 hr into the light phase (time of PET scan) for 1 week. The day before PET/CT scans, mice were fasted overnight (18 hr) with ad lib access to water. On the experimental day, mice were injected IP with 10 mg/kg Benserazide (Selleckchem.com) and 10 mg/kg Entacapone (Selleckchem.com) 30 min before injecting radiolabelled fDOPA (approx. 5 MBq) into the tail vein. After 5 min rest, mice received one peanut butter chip and only those mice that initiated feeding within the first minute were included in the analysis. Then mice were anaesthetised (1–2.5% isoflurane) and a 65 min PET/CT scan was performed under anaesthesia. The scan was acquired using the Inveon scanner (Siemens Inveon). CT was generated using the following parameters: 97 μm of resolution, 80 kV voltage and 500 μA current, mainly

for attenuation correction and MRI overlaying purposes. PET scans were acquired for 60 min (16 time frames). One week later MRI scan was performed on these animals to generate T2 weighted images using these parameters: 3D Flash sequence, TE/TR = 8/60ms, 4 averages, flip angle = 10 degree, resolution = 0.155 mm$^3$, scan time 20 min. PET images were analysed using IRW software (Inveon Research Workplace 4.2). PET/CT images were overlaid manually with the T2-Weighted images. Region of Interests (ROIs) were generated carefully by one investigator (AR) on the T2 images on left/right ventral striatum, left/right dorsal striatum and cerebellum (according to Allen brain atlas) and PET voxel intensities (Unit Bq/ml) at three different time points were exported for further analysis. We specifically choose to use the ventral striatum, rather than the NAc, due to greater accuracy with ROI identification. SUV (Standardised Uptake Value) of ROIs were calculated for each time point using the following equation: SUV = $C_{PET}(T)$/(Dose/Weight). $C_{PET}(T)$ = Tissue radioactivity concentration at time T.

Dose = administration dose at time of injection (decay corrected to the time points). Weight = animal body weight. To eliminate bias for slight variation of size/location of ROI we used upper bound Bq/ml values to calculate SUVmax (*Kinahan and Fletcher, 2010*) and normalised to cerebellum as reference:

SUVRmax = SUVmax target/SUVmax reference.

## Immunohistochemistry

To confirm viral injection and fiber placements, animals were deeply anaesthetised with isoflurane and perfused with 0.05 M PBS, followed by 4% paraformaldehyde. Brains were postfixed in 4% paraformaldehyde overnight at 4 °C then placed in 30% sucrose. Brains were cut at 40 µm on a cryostat, and every fourth section was collected and stored in cryoprotectant at −20 °C. Sections were washed in 0.1 M phosphate buffer (PB), incubated with 1% hydrogen peroxide ($H_2O_2$) for 15 minutes to prevent endogenous peroxidase, activity, and blocked for 1 hr with 5% normal horse serum (NHS) in 0.3% Triton in 0.1 M PB. Sections were incubated with chicken anti-GFP (ab13970, Abcam) at 1:1000 in diluent of 5% NHS in 0.3% Triton in 0.1 M PB. After incubation, the sections were washed and incubated with Alexa Fluor goat anti-chicken 488 antibody (Invitrogen) at 1:500 in 0.3% Triton in 0.1 M PB. Sections were then washed, mounted, and coverslipped.

## Statistical analysis

Statistical analyses were performed using GraphPad Prism for MacOS X. Data are represented as mean ± SEM. Two-way ANOVAs with post hoc tests were used to determine statistical significance. A two-tailed Student's paired or unpaired t-test (see figure legends for specific details) was used when comparing genotype only. < 0.05 was considered statistically significant and is indicated on figures and in figure legends.

## Acknowledgements

We would like to thank Myles Billard from TDT for his valuable technical assistance and support with setting up photometry and analysis. We would like to thank Antonio (Nino) Benci at the Monash Instrumentation Facility for help with FED3 production and maintenance. The authors acknowledge the facilities and scientific and technical assistance of the National Imaging Facility, a National Collaborative Research Infrastructure Strategy (NCRIS) capability, at Monash Biomedical Imaging, Monash University. We would like to thank Professor Alex Fornito for the use of fDOPA in PET studies. We acknowledge that Bio Render was used to produce elements incorporated in the figure and graphical abstract (Biorender.com). Funding: National Health and Medical Research Council project grant APP1126724 (ZBA). National Health and Medical Research Council research fellowship APP1154974 (ZBA).

## Additional information

### Funding

| Funder | Grant reference number | Author |
|---|---|---|
| National Health and Medical Research Council | 1126724 | Zane B Andrews |
| National Health and Medical Research Council | 1154974 | Zane B Andrews |

The funders had no role in study design, data collection and interpretation, or the decision to submit the work for publication.

### Author contributions

Alex Reichenbach, Conceptualization, Data curation, Formal analysis, Investigation, Methodology, Resources, Software, Validation, Visualization, Writing - original draft, Writing – review and editing; Rachel E Clarke, Formal analysis, Investigation, Methodology, Visualization; Romana Stark, Sarah Haas Lockie, Tara Sepehrizadeh, Formal analysis, Investigation, Methodology; Mathieu Mequinion, Sasha Rawlinson, Felicia Reed, Investigation, Methodology; Harry Dempsey, Formal analysis, Investigation, Methodology, Software; Michael DeVeer, Data curation, Formal analysis, Investigation, Methodology, Supervision, Writing – review and editing; Astrid C Munder, Data curation, Formal analysis, Investigation, Methodology, Resources, Supervision, Writing – review and editing; Juan Nunez-Iglesias, Formal analysis, Investigation, Methodology, Software, Supervision; David C Spanswick, Data curation, Formal analysis, Investigation, Methodology, Resources, Software, Supervision, Writing – review and editing; Randall Mynatt, Resources; Alexxai V Kravitz, Resources, Software, Validation, Visualization; Christopher V Dayas, Conceptualization, Investigation, Methodology, Writing – review and editing; Robyn Brown, Conceptualization, Investigation, Methodology, Resources, Supervision, Writing – review and editing; Zane B Andrews, Conceptualization, Data curation, Formal analysis, Funding acquisition, Investigation, Project administration, Resources, Supervision, Validation, Visualization, Writing - original draft, Writing – review and editing

### Author ORCIDs

Alex Reichenbach (iD) http://orcid.org/0000-0002-3520-8341
Harry Dempsey (iD) http://orcid.org/0000-0001-5117-6995
Alexxai V Kravitz (iD) http://orcid.org/0000-0001-5983-0218
Zane B Andrews (iD) http://orcid.org/0000-0002-9097-7944

### Ethics

All experiments were conducted in compliance with the Monash University Animal Ethics Committee guidelines (MARP 17855).

### Decision letter and Author response

Decision letter https://doi.org/10.7554/eLife.72668.sa1
Author response https://doi.org/10.7554/eLife.72668.sa2

## Additional files

### Supplementary files

• Transparent reporting form

### Data availability

All data generated or analysed during this study are included in the manuscript and supporting files. Source data files have been provided for Figures 1-6, Figure 1-figure supplements 1&2, Figure 6-figure supplement 1.

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
