## [Editor Report]

This manuscript will be of broad interest to behavioral neuroscientists studying energy homeostasis, hypothalamic feeding circuits, and dopamine. The paper uses a genetic mouse model to study critical connections between homeostatic circuitry and dopamine release in response to food reward. The experimental results support key claims of the paper, and tie in nicely with previously published data.

---

## [Decision Letter]

**Decision letter after peer review:**

Thank you for submitting your article "Metabolic sensing in AgRP neurons integrates homeostatic state with dopamine signalling in the striatum" for consideration by *eLife*. Your article has been reviewed by 3 peer reviewers, including Richard D Palmiter as Reviewing Editor and Reviewer #1, and the evaluation has been overseen by Gary Westbrook as the Senior Editor. The following individual involved in review of your submission has agreed to reveal their identity: Herbert Herzog (Reviewer #3).

Essential revisions:

1. We assume that after eating a PB chip the inhibitory input to AgRP neurons from DMH (and elsewhere) increases leading the reduced fiber photometry readings in AgRP neurons. With consumption of more chips the inhibition increases, but the AgRP neurons in KO mice cannot respond more due to metabolic-sensing defect. But, what if chronic Crat deficiency in AgRP neurons affects the activity of neurons that synapse onto AgRP neurons due to compensatory changes. For example, maybe inhibitory DMH (or other) neurons become less response to signals from the gut. Thus, we suggest that the authors compare mini IPSCs of WT and KO mice before and after consuming a few PB chips, They might also assess evoked IPSCs by stimulating the DMH and recording from AgRP neurons before and after chip consumption.

2. There are also suggestions below for improving the display and analysis of the data that should be given serious consideration.

*Reviewer #1 (Recommendations for the authors):*

1. Fasting enhances the motivation of rodents to seek and work for food. Dopamine release in the striatum, especially in the nucleus accumbens, underlies the motivational response to hunger. The authors of this paper provide genetic evidence that metabolic sensing by hypothalamic AgRP neurons contributes to the enhanced dopamine signaling in the striatum. The suppression of AgRP-neuron activity with repeated consumption of palatable chips in Crat-KO mice is postulated to be due to inadequate metabolic sensing by AgRP neurons i.e., the neurons cannot respond to further inhibitory input that follows food consumption in the absence of Crat. If metabolic sensing is an intrinsic property of AgRP neurons, then it should be independent of input. Thus, it is important to show that inhibitory input to AgRP neurons is unaltered when comparing WT and KO mice after consumption of several PB chips.

2. In Figure 1J, it would be useful to know the times of PB chip consumption when these measurements were made. Figure 1E,F suggest that KO neurons are less responsive to glucose. Are the measurements made before or after there is a measurable change in circulating glucose levels?

3. Discussion of how lack of Crat might limit the ability of AgRP neurons to respond normally to inhibitory inputs should be added.

*Reviewer #2 (Recommendations for the authors):*

– I think simplest explanation for the abnormal response of AgRP neurons to food presentation and the decrease in DA release to palatable rewards may be that AgRP neurons are less active at baseline and/or less excitable by fasting in KO mice. As noted above, the relatively normal ex vivo measurements may not accurately reflect this. Additional work to directly and indirectly address this possibility may include further slice studies (comparing fasted and fed KO and control mice), additional fiber photometry studies to assess baseline AgRP neuron activity over time (analogous to Mazzone et al., 2020), and optogenetics studies to assess whether exogenous stimulation of AgRP neurons can overcome deficits in motivated behavior observed in the knockout mice. I do not necessarily think that this paper needs to answer this question with additional experiments; however, if already done or underway these data would add substantially to the paper. Barring that, addressing this possibility/avenue for future experiments in the discussion would be adequate from my perspective

– In figure 1 – food intake/re-feeding data to accompany the photometry experiments showing decreased AgRP neuron responses to repeat rewards should be presented to show the correlation between the AgRP neuron response to food and subsequent food intake. Prior studies by the authors have showed decreased re-feeding in KO mice over 24 hours but have not reported food intake immediately following food presentation. If immediate food intake does not correlate with AgRP neuron inhibition on food presentation/re-presentation it would be the first example where these findings are dissociated and would warrant explanation and follow-up.

– For the experiments in figure 4 (and other GRAB-DA experiments), how are mice evaluated for inclusion or exclusion from photometry experiments. Photometry signals between animals can vary much more than within animals, and comparing groups of 4 WT vs KO mice as is done in D and E could yield skewed results based on low virus expression or implant placement (and the latter can be challenging to determine precisely post-mortem).

– Related to this, to more accurately represent the biological data collected, I think the scatter plots and statistics in figure 4 F,G,J should use the number of mice as N and average nose poke z-scores within animal. If I'm incorrect about this, would appreciate an explanation.

– In figure 6, I think – as above – that z-scores should be presented by animal in the scatter plots in F,I,L,M. I could understand using z-scores from individual nose pokes if trying to show a difference in DA response across the session (or something like that), but since this is not the point of the figure I think treating nose pokes within animal as technical replicates and each animal as a biological replicate with a single data point on the scatter plot is appropriate.

Recommendations for improving the writing and presentation.

– the authors describe how AgRP photometry implants are determined to be functional with ghrelin injection. What were the cut-offs for inclusion? It would also be helpful to know how GRAB-DA implants are evaluated for inclusion either prior to experiments or post-mortem.

– why is sex mentioned in some but not all figure legends. Where not mentioned are they all male or all female?

– it would be helpful to mention in figure legends or schematics where PB chips are used whether mice in a given experiment are naïve or habituated to the chips.

Correction to text and figures:

– figure supplement 1 – I think A/ B and C/D are reversed on the legend relative to the figure.

– figure 4 – does panel J contain any data not in F,G? If not I think it can be omitted. Apologies if I'm missing something.

– figure 5 – D,E,H,I – Y axes seem pretty expanded. Responses to all stimuli would be easier to appreciate if they could be scaled down.

– line 166 – Beutler et al., 2017 also show that the response to presentation of food predicts calories consumed using complementary experiments and should be cited alongside Su et al., 2017 when discussing this point.

*Reviewer #3 (Recommendations for the authors):*

Almost all data are presented as Z scores. While this is acceptable it does not allow to get an idea of the magnitude of the activity change. Is there some other metric that could be used at least in one experiment to allow this to be compared to other responses seen in the literature?

Despite missing the mention of (I,J,K,L) in the legend of SubFigure 2, the data on the behavioural experiments (L/D box and EPM) need to be extended showing all parameters eg #entry into the different zones, total distance travel in the zone and over all, etc to allow to give a conclusive answer whether the animals show anxiety behaviour or not.

The discussion is rather long and a bit repetitive, it could be more concise and only fucus on the main points. It could however, expand a bit on how the authors think AgRP neurons mediate this response to dopamine changes eg is it dependent on AgRP, NPY or GABA signalling from these neurons etc? Are AgRP neurons the only ones that mediate energy sensing responses eg what is the role of POMC neurons in that process?

In the PET analysis were any other brain areas also affected and analysed by the treatment?

---

## [Author Response]

Reviewer #1 (Recommendations for the authors):1. Fasting enhances the motivation of rodents to seek and work for food. Dopamine release in the striatum, especially in the nucleus accumbens, underlies the motivational response to hunger. The authors of this paper provide genetic evidence that metabolic sensing by hypothalamic AgRP neurons contributes to the enhanced dopamine signaling in the striatum. The suppression of AgRP-neuron activity with repeated consumption of palatable chips in Crat-KO mice is postulated to be due to inadequate metabolic sensing by AgRP neurons i.e., the neurons cannot respond to further inhibitory input that follows food consumption in the absence of Crat. If metabolic sensing is an intrinsic property of AgRP neurons, then it should be independent of input. Thus, it is important to show that inhibitory input to AgRP neurons is unaltered when comparing WT and KO mice after consumption of several PB chips.

We would like to thank the reviewer for this important suggestion, however due to the ongoing pandemic lockdowns in Victoria Australia, we have been unable to complete these ephys experiments. We have lacked both expertise and resources to complete these studies with our ephys expert unable to enter Victoria Australia for most of the revision period.

Nevertheless, we wanted to address this issue by providing additional data to support our conclusions that impaired dopamine signaling in AgRP Crat KO mice was a failure of metabolic sensing. To do this, we looked at in vivo AgRP neural responses to an IP glucose challenge, which is known to suppress AgRP neural activity (3). Although IP glucose injection suppressed AgRP activity in WT and KO mice, activity quickly returned to baseline levels in fasted WT but not KO mice (new data Figure 1—figure supplement 1J-M). Thus, in fasted WT mice calorie availability was insufficient to sustain the suppression of AgRP neural activity and activity returned to pre-glucose levels to encourage further food seeking. Although fasted KO mice reduced activity in response to IP glucose, AgRP activity did not return to pre glucose levels indicating an inability to sense further calorie need. With the addition of this new data, we show that AgRP crat deletion affects both ex vivo and in vivo AgRP neural responses to glucose, as well as in response to palatable food.

Furthermore, the majority of inhibitory inputs into AgRP neurons come from leptin-receptor expressing DMH (DMH^LEPR^) GABA neurons (4) and it was recently demonstrated that DMH^LEPR^ neurons provide inhibitory input to AgRP neurons in response to sensory cues but with only a minimal response to metabolic signals (5). Thus, we feel our data, together with evidence from the literature, strongly supports the conclusion that impaired metabolic sensing in AgRP neurons, and not alterations in inhibitory inputs, is responsible for the changes in dopamine signalling observed.

We have added more detailed and addressed this as a limitation (below) in the discussion.

“It should be noted, however, that we cannot rule out the possibility of reduced synaptic inhibitory input on to AgRP neurons in KO mice after initial PB presentation. The majority of inhibitory input to AgRP neurons comes from GABAergic leptin-receptor expressing DMH (DMH^LEPR^) neurons (4). Importantly, DMH^LEPR^ neurons provide inhibitory input to AgRP neurons in response to sensory cues but with only a minimal response to metabolic signals (5), arguing against the possibility of altered inhibitory input in AgRP neurons from KO mice in our study. It is also possible that Crat deletion from AgRP neurons alters the post-synaptic response to GABAergic inhibitory inputs since altered mitochondrial glucose metabolism, as observed after crat deletion, affects GABA metabolism(6)”.

2. In Figure 1J, it would be useful to know the times of PB chip consumption when these measurements were made.

PB recording were made in the light phase. This has been added to the methods sections. There were no differences between WT and KO mice, as can be seen in Author response image 1. We have not added the follow recording time data to the manuscript but are happy to provide here in rebuttal document.

**Author response image 1. sa2fig1:** 

Added

“Recordings were conducted during the light phase, with the majority towards the end of the light phase”.

Figure 1E,F suggest that KO neurons are less responsive to glucose. Are the measurements made before or after there is a measurable change in circulating glucose levels?

This is an excellent point. In terms of in vivo changes in AgRP activity, no differences were observed in response to PB when presented for the first time, however, subsequent exposures showed significantly greater fall in activity in WT and but not KO mice. Thus, the integration of available caloric information (assumed to included changes in plasma glucose) must come after the first exposure. Subsequent exposure to PB in both WT and KO was conducted over multiple days, so all mice would have been available to integrate metabolic after first exposure. However, changes in activity to subsequent exposure occur immediate as PB is dropped into the cage (see videos 3-6) and these changes occur prior to any measurable difference in plasma glucose.

In addition, GRAB-DA responses occur prior to measurable changes in plasma glucose, but again all mice have been habituated to PB pellets for multiple days prior to recordings. It is interesting to note that fDOPA PET/CT data only show changes in the dorsal striatum 30 mins after PB pellet exposure, during which time measurable changes in blood glucose can be observed. In fact, our previous studies show impaired glucose clearance in KO mice 15 mins after a glucose challenge (1), further supporting the inability to sense plasma glucose over a time frame similar to PET/CT studies.

3. Discussion of how lack of Crat might limit the ability of AgRP neurons to respond normally to inhibitory inputs should be added.

Added.

“It should be noted, however, that we cannot rule out the possibility of reduced synaptic inhibitory input on to AgRP neurons in KO mice after initial PB presentation. The majority of inhibitory input to AgRP neurons comes from GABAergic leptin-receptor expressing DMH (DMH^LEPR^) neurons (4). Importantly, DMH^LEPR^ neurons provide inhibitory input to AgRP neurons in response to sensory cues but with only a minimal response to metabolic signals (5), arguing against the possibility of altered inhibitory input in AgRP neurons from KO mice in our study. It is also possible that Crat deletion from AgRP neurons alters the post-synaptic response to GABAergic inhibitory inputs since altered mitochondrial glucose metabolism, as observed after crat deletion, affects GABA metabolism(6)”.

Reviewer #2 (Recommendations for the authors):– I think simplest explanation for the abnormal response of AgRP neurons to food presentation and the decrease in DA release to palatable rewards may be that AgRP neurons are less active at baseline and/or less excitable by fasting in KO mice. As noted above, the relatively normal ex vivo measurements may not accurately reflect this. Additional work to directly and indirectly address this possibility may include further slice studies (comparing fasted and fed KO and control mice), additional fiber photometry studies to assess baseline AgRP neuron activity over time (analogous to Mazzone et al., 2020), and optogenetics studies to assess whether exogenous stimulation of AgRP neurons can overcome deficits in motivated behavior observed in the knockout mice. I do not necessarily think that this paper needs to answer this question with additional experiments; however, if already done or underway these data would add substantially to the paper. Barring that, addressing this possibility/avenue for future experiments in the discussion would be adequate from my perspective

Differences in baseline fluorescent signal is a problem for all calcium imaging approaches and this is reflected in several variables that will always be slightly different for each mouse. These including (1) GCaMP expression, (2) irradiance, (3) light intensity within tissue, (4) quality of surgery (ie gliosis), (5) position of fibre optic implant. We make every attempt to minimise the impact of these variables and described in the methods how we try to mitigate this and described this in the methods.

From methods

”Prior to experiments, baseline GCaMP6s or dopamine signal was measured and LED power was adjusted for each mouse to achieve approximately 200 mV for 465 nm (530 Hz) and 100 mV for 405 nm (210 Hz). 405 nm was used as an isosbestic control, which is wavelength at which GCaMP6 excitation is independent from intracellular [ca^2+^], allowing for an assessment of motion artefact. This approach is designed to minimise the impact of variable GCaMP6s expression, tissue irradiance, surgery quality and distance of the fibre optic from GCaMP6s”.

We have also added the following text to the discussion.

“One potential limitation from our studies is that reduced dopamine release in KO mice may reflect less active AgRP neurons at baseline or during fasting. Although our ex vivo electrophysiology studies show no differences at baseline, they cannot rule out in vivo differences that could not be addressed with ex vivo slice recording or population calcium changes using GCaMP6”.

– In figure 1 – food intake/re-feeding data to accompany the photometry experiments showing decreased AgRP neuron responses to repeat rewards should be presented to show the correlation between the AgRP neuron response to food and subsequent food intake. Prior studies by the authors have showed decreased re-feeding in KO mice over 24 hours but have not reported food intake immediately following food presentation. If immediate food intake does not correlate with AgRP neuron inhibition on food presentation/re-presentation it would be the first example where these findings are dissociated and would warrant explanation and follow-up.

We designed these experiments so that mice were given single PB pellets, weighing approximately 70 mg, and all was consumed during exposure – therefore all mice ate the approximate amount. We have now described this in the methods.

“Peanut butter chips were measured to ~70mg per pellet and one pellet was given per trial. Mice consumed all of this peanut butter chip during each trial such that no differences in consumption were observed.”

We have also added the additional data.

During each trial the time to peanut butter consumption was not different between genotypes (new data - Figure 1-figure supplement 1G,H,I)

– For the experiments in figure 4 (and other GRAB-DA experiments), how are mice evaluated for inclusion or exclusion from photometry experiments. Photometry signals between animals can vary much more than within animals, and comparing groups of 4 WT vs KO mice as is done in D and E could yield skewed results based on low virus expression or implant placement (and the latter can be challenging to determine precisely post-mortem).

The increases in dopamine release before contact with PB in the NAc or dorsal striatum (Figure 2L-P; Figure 5L,M) and before pellet retrieval (in Figure 4K-N; Figure 6 J, K) is similar between genotypes, suggesting equivalent capacity to increase dopamine release to stimuli not affected by consumption. Thus, genotype differences in response to palatable food or sucrose pellets could not be due to differences in GRAB-DA expression in the NAc. Moreover, a postmortem analysis was conducted to identify the localization of GFP expression (Figure 7).

We have added the following text.

“GRAB-DA responses in WT and KO mice were similar on approach to PB and prior to pellet retrieval in both the NAc and dorsal striatum showing that genotype differences in response to palatable food or sucrose pellets could not be due to differences in GRAB-DA expression in the NAc. Moreover, a post mortem analysis was conducted to identify the localization of GFP expression (Figure 7)”.

– Related to this, to more accurately represent the biological data collected, I think the scatter plots and statistics in figure 4 F,G,J should use the number of mice as N and average nose poke z-scores within animal. If I'm incorrect about this, would appreciate an explanation.– In figure 6, I think – as above – that z-scores should be presented by animal in the scatter plots in F,I,L,M. I could understand using z-scores from individual nose pokes if trying to show a difference in DA response across the session (or something like that), but since this is not the point of the figure I think treating nose pokes within animal as technical replicates and each animal as a biological replicate with a single data point on the scatter plot is appropriate.

We also recognised this as a weakness and have now repeated these experiments using multiple additional animals. All operant GRAB-DA data in the accumbens and dorsal striatum is now presented as the averaged nose poke or pellet responses recorded for each mouse. Data points in Figure 4 and Figure 6 now reflect analysis by mouse.

We have also updated this in the methods section:

“For DA photometry during PR session, data are presented as the averaged dopamine response for each animal. On average mice collected ~3.5 pellets during the PR in ad libitum fed conditions and ~6 pellets when fasted (Figure 4O).”

Recommendations for improving the writing and presentation.– the authors describe how AgRP photometry implants are determined to be functional with ghrelin injection. What were the cut-offs for inclusion? It would also be helpful to know how GRAB-DA implants are evaluated for inclusion either prior to experiments or post-mortem.

For AgRP photometry studies, only mice with a maximal z-score response to IP ghrelin greater than 4 were included for analysis, ensuring differences in AgRP neural activity to food cues were not related to differences in GCaMP expression or illumination rates.

We have added the following text to the manuscript.

“For AgRP GCaMP6 photometry studies, an increase in activity to ghrelin response was used as an index of correct viral expression and fibre optic placement. Only mice with a maximal peak z-score of >4 were included for analysis in experimental group, using this criterion 5/20 mice, across both WT and KO mice, were excluded for experimentation (Figure 1-figure supplement 1N-O)”.

In addition, increases in dopamine release before contact with PB in the NAc or dorsal striatum (Fig 2L-P; Fig 5L,M) and before pellet retrieval (in Fig 4K-N; Fig 6 J, K) is similar between genotypes, suggesting equivalent capacity to increase dopamine release under stimuli not affected by AgRP input. Thus, genotype difference in response to palatable food or sucrose pellets could not be due to differences in GRAB-DA expression in the NAc. Moreover, a postmortem analysis was conducted to identify the localization of GFP expression (Figure 7).

We have added the following text.

“GRAB-DA responses in WT and KO mice were similar on approach to PB and prior to pellet retrieval in both the NAc and dorsal striatum showing that genotype differences in response to palatable food or sucrose pellets could not be due to differences in GRAB-DA expression in the NAc. Moreover, a post mortem analysis was conducted to identify the localization of GFP expression (Figure 7)”.

– why is sex mentioned in some but not all figure legends. Where not mentioned are they all male or all female?

Not all studies involved females – females were used when mentioned in figure legends.

– it would be helpful to mention in figure legends or schematics where PB chips are used whether mice in a given experiment are naïve or habituated to the chips.

Only in Figure 1 were mice naïve to peanut butter. Naïve mice in experiment 1 took between 1 and 5 min before approaching peanut butter chip. For our experimental design in Figure 2 and 5 we needed guaranteed approach and consumption within a reasonable timeframe, we therefore provided 1 peanut butter pellet daily for 5 days to accustom mice to peanut butter

We have included the following text in the methods.

“To measure AgRP responsiveness to palatable food, mice were presented with Reese’s peanut butter chips. Mice were habituated to peanut butter chips for 5 days (1x 70mg pellet/day) unless otherwise stated as naïve (Figure 1G-H only)”.

Correction to text and figures:

Thank you for pointing these out.

– figure supplement 1 – I think A/ B and C/D are reversed on the legend relative to the figure.

changed

– figure 4 – does panel J contain any data not in F,G? If not I think it can be omitted.

Removed and added a graph (Figure 4O) showing the pellets collected by each mouse during PR session (as this can’t be seen in the averaged data for each mouse).

Apologies if I'm missing something.– figure 5 – D,E,H,I – Y axes seem pretty expanded. Responses to all stimuli would be easier to appreciate if they could be scaled down.

done

– line 166 – Beutler et al., 2017 also show that the response to presentation of food predicts calories consumed using complementary experiments and should be cited alongside Su et al., 2017 when discussing this point.

Reference included.

Reviewer #3 (Recommendations for the authors):Almost all data are presented as Z scores. While this is acceptable it does not allow to get an idea of the magnitude of the activity change. Is there some other metric that could be used at least in one experiment to allow this to be compared to other responses seen in the literature?

A normalisation approach is commonly used in the field. This allows comparison to a defined baseline where a z-score of 1 represents 1 standard deviation away for the baseline and z-score 2 represents 2 standard deviations away from the baseline etc.

We have now also included supplement videos of real-time changes in GCaMP6s or GRAB-DA fluoscence (Videos 1-6).

Despite missing the mention of (I,J,K,L) in the legend of SubFigure 2, the data on the behavioural experiments (L/D box and EPM) need to be extended showing all parameters eg #entry into the different zones, total distance travel in the zone and over all, etc to allow to give a conclusive answer whether the animals show anxiety behaviour or not.

We have now added this new data in Figure 1—figure supplement 2I-P. There is no difference in open arm time or entries in the elevated plus maze and no differences in the time spent in the light zone of light zone entries in the light dark box. There were no differences in the light zone velocity and distance moved however, KO mice moved less with a reduced velocity in in the ad libitum fed state.

The discussion is rather long and a bit repetitive, it could be more concise and only fucus on the main points. It could however, expand a bit on how the authors think AgRP neurons mediate this response to dopamine changes eg is it dependent on AgRP, NPY or GABA signalling from these neurons etc? Are AgRP neurons the only ones that mediate energy sensing responses eg what is the role of POMC neurons in that process?

We have now modified the discussion to shorten.

In the PET analysis were any other brain areas also affected and analysed by the treatment?

No other regions were examined. Regions had to be manually annotated, which was very time consuming. The striatum is the major region for accumulation of fDOPA and our hypothesis was specific to the striatum (Ventral Striatum [NAc] and dorsal striatum).